**EMBO** *reports*

# Bora bridges Aurora-A activation and substrate recognition of PLK1

Jennifer A Miles[1,2], Matthew Batchelor [1,2], Martin Walko [2,3], Vanda Gunning [1,2], Andrew J Wilson [2,3,4], Megan H Wright[2,3] & Richard Bayliss [1,2]✉

## Abstract

The activation of PLK1 in late G2 is critical for mitotic entry, requiring its phosphorylation by Aurora-A, facilitated by the intrinsically disordered protein Bora. The structural basis of this mechanism has remained unresolved. Here, we present models of the Aurora-A/Bora complex and the Aurora-A/Bora/PLK1 complex, validated with site-specific mutagenesis, biochemical assays and NMR spectroscopy. Bora wraps around the N-lobe of Aurora-A, occupying the pockets used by its other activators. A CDK1 phosphorylation site on Bora (Ser112) mimics the structural role of Aurora-A activation loop phosphorylation within a TPX2-like binding motif. In the ternary complex, Bora bridges the two kinases, orienting the activation loop of PLK1 towards the active site of Aurora-A. Bora residues 56–66 form a critical interface with a conserved pocket on the PLK1 C-helix that is analogous to the TPX2-binding Y-pocket of Aurora-A. Aurora-A phosphorylation of Bora Ser59 creates an additional interaction that increases the efficiency of PLK1 phosphorylation. These findings deepen our understanding of Aurora-A regulation by its disordered binding partners and establish a mechanistic framework for Bora-dependent activation of PLK1.

Keywords Bora; Aurora-A; PLK1; Protein Kinase; Phosphorylation
Subject Categories Cell Cycle; Post-translational Modifications & Proteolysis; Structural Biology

See also: M Gobran & P Lenart, A Esposito-Verza et al and A Pillan et al

## Introduction

Protein kinases such as Aurora-A, PLK1 and CDK1 have critical roles in orchestrating mitosis, especially in the regulation of the G2/M transition and mitotic spindle assembly and function. Aurora-A is a member of the Aurora family of kinases, that also includes kinetochore-associated Aurora-B and Aurora-C. It has roles in

mitotic spindle assembly, DNA repair, centrosome maturation and cilia regulation (Willems et al, 2018). Structurally, Aurora-A contains a disordered N-terminal domain (aa 1–121), followed by a kinase domain (aa 122–387) and a short, disordered C-terminal region (aa 388–403). Canonical activation of its kinase function requires phosphorylation of Thr288 in the activation loop, a process that primarily occurs by autophosphorylation. Phosphory-lated Aurora-A is enriched on spindle poles and is also found along spindle microtubules close to the poles (Ohashi et al, 2006; Holder et al, 2024). Aurora-A activity is tightly regulated by activators and inhibitors, in particular through the binding of intrinsically disordered proteins that 'complete' the incomplete Aurora-A core kinase domain (Levinson, 2018; Bayliss et al, 2012). Activator proteins, such as TPX2 (Bayliss et al, 2003), can stimulate Aurora-A autophosphorylation on Thr288, leading to an active kinase.

PLK1 is a member of the Polo-like kinase family first identified in *Drosophila melanogaster* which is comprised of 5 members in humans (Llamazares et al, 1991; Sunkel and Glover, 1988; Korns et al, 2022). PLK1 is composed of an N-terminal kinase domain linked to a polo-box domain (PBD) comprised of two polo-boxes and an upstream linker (Elia et al, 2003; Cheng et al, 2003). The PBD is required for recognition of protein substrates and control of PLK1 localisation (Park et al, 2010). PLK1 is frequently over-expressed in cancer and is linked to a poor prognosis (Eckerdt et al, 2005). Activation of PLK1 is achieved through the phosphorylation of Thr210 by Aurora-A kinase (Macůrek et al, 2008; Seki et al, 2008a; Chan et al, 2008) during the G2/M transition following successful DNA damage repair (Macůrek et al, 2008). Active PLK1 then phosphorylates and activates key mitotic regulators such as the phosphatase CDC25 and the ubiquitin ligase APC/C (Moshe et al, 2004; Qian et al, 2001).

Bora is an intrinsically disordered protein of 559 amino acids that was discovered in *Drosophila melanogaster* (Hutterer et al, 2006) and its overexpression has been observed in human bladder and colorectal cancer, and adenocarcinoma samples (Cheng et al, 2020; Mahajan et al, 2023; Zhang et al, 2017). The selective modification of PLK1 by Aurora-A on Thr210 is mediated by Bora (Seki et al, 2008b; Macůrek et al, 2008; Hutterer et al, 2006). Bora has a high content of serine (15%) and threonine (6.6%) residues (Thomas et al, 2016) and a total of fourteen sites are phosphory-lated by CDK1–Cyclin A in vitro and twelve in vivo (Feine et al,

[1]School of Molecular and Cellular Biology, Faculty of Biological Sciences, University of Leeds, Leeds LS2 9JT, UK. [2]Astbury Centre for Structural Molecular Biology, University of Leeds, Leeds LS2 9JT, UK. [3]School of Chemistry, Faculty of Engineering and Physical Sciences, University of Leeds, Leeds LS2 9JT, UK. [4]School of Chemistry, University of Birmingham, Birmingham B15 2TT, UK. ✉E-mail: r.w.bayliss@leeds.ac.uk

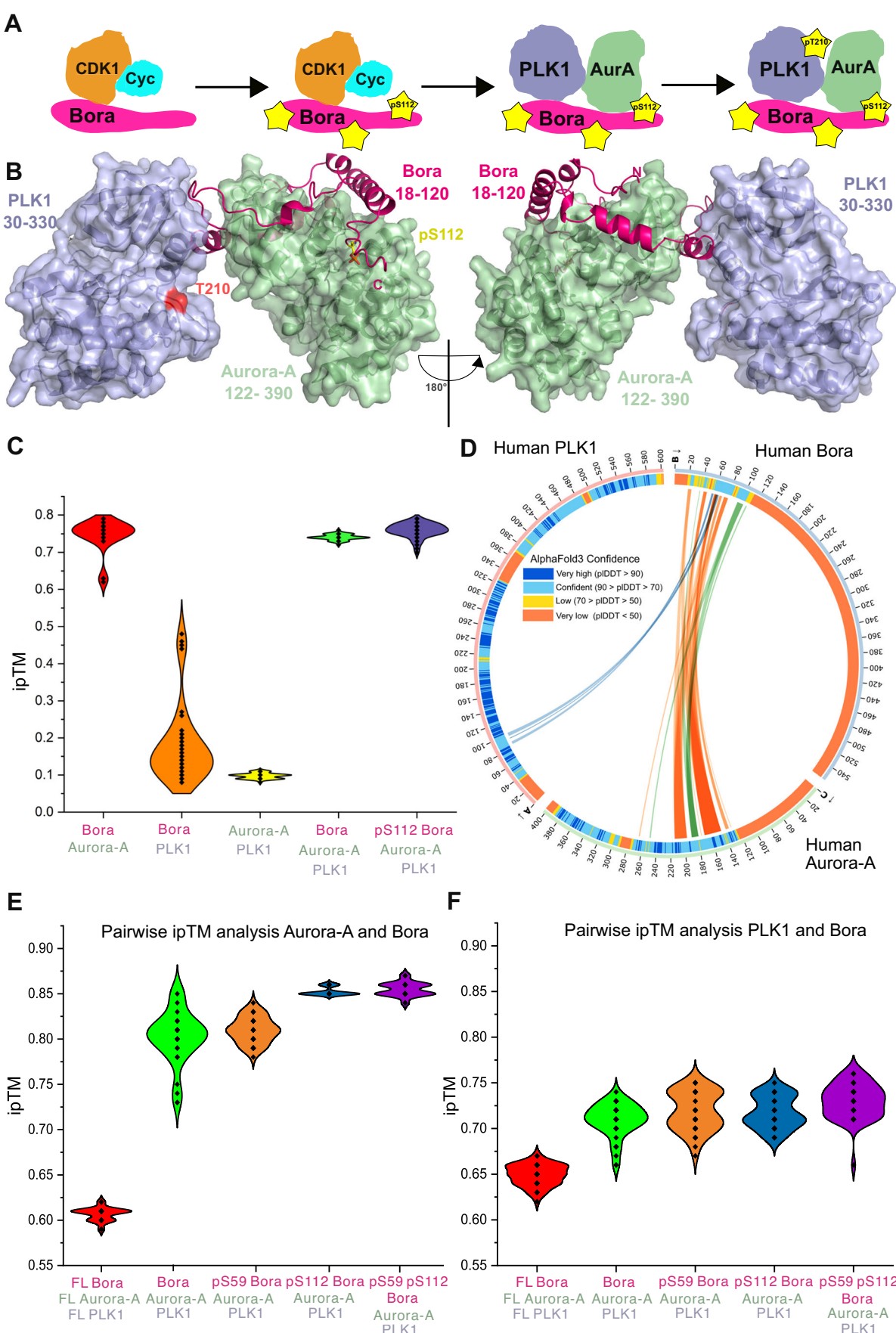

◄ **Figure 1. Bora bridges Aurora-A activation and substrate recognition of PLK1.**

(A) Model showing the role of Bora in the facilitation of PLK1 phosphorylation by Aurora-A kinase. Bora is phosphorylated on numerous sites, including Ser112 by CDK1–Cyclin A. Phosphorylated Bora then binds more tightly to Aurora-A and acts in *trans* to activate the unphosphorylated Aurora-A, resulting in the phosphorylation of PLK1 at Thr210 (human numbering). (B) AlphaFold3 model of the truncated ternary complex between human Aurora-A, PLK1 and Bora. Aurora-A 122–390 is shown in green, with PLK1 30–330 in blue and Bora 18–120 in magenta. Thr210 in PLK1 that is modified by Aurora-A is highlighted in red, with the phosphorylated Ser112 in human Bora highlighted in yellow. (C) Violin plot of iPTM scores from 50 models of the truncated binary and ternary complexes (Bora 18–120, Aurora-A 122–403 and PLK1 30–330) produced by AlphaFold3 (using alphafoldserver.com). $n = 50$. (D) AlphaBridge representation of the ternary complex between human Aurora-A, PLK1 and Bora. The sequences around the outside are coloured based on the confidence in the modelling of these regions, with blue indicating high confidence and orange indicating low confidence. The residues predicted to be interacting between the three proteins are linked with curved lines, with blue indicating the interface between PLK1 and Bora, and orange/green indicating residues at the different interfaces between Aurora-A and Bora. (E) Violin plot of iPTM scores between Bora and Aurora-A in the ternary complexes of Bora, Aurora-A and PLK1 produced by AlphaFold3 (using alphafoldserver.com). The phosphorylation state of Bora is listed. FL = full-length, with all the other complexes being modelled with the truncated proteins (Bora 18–120, Aurora-A 122–403 and PLK1 30–330). $n = 10$. (F) Violin plot of iPTM scores between Bora and PLK1 in the ternary complexes of Bora, Aurora-A and PLK1 produced by AlphaFold3 (using alphafoldserver.com). The phosphorylation state of Bora is listed. FL = full-length, with all the other complexes being modelled with the truncated proteins (Bora 18–120, Aurora-A 122–403 and PLK1 30–330). $n = 10$. Source data are available online for this figure.

2014; Thomas et al, 2016). Three of the sites that are modified by CDK1–Cyclin A (Ser41, Ser112, Ser137) are evolutionarily conserved and important for the function of Bora, including in *C. elegans* as well as human cells (Thomas et al, 2016; Tavernier et al, 2015; Parrilla et al, 2016; Pintard and Archambault, 2018).

Phosphorylation of Bora at three N-terminal phosphorylation sites (S41, S112, S137) promotes the phosphorylation of PLK1 on Thr210 (Thomas et al, 2016). The roles of other phosphorylation sites have yet to be defined, although Thr52 phosphorylation may be required for degradation of Bora (Feine et al, 2014).

The precise mechanism by which PLK1, Aurora-A and Bora come together to bring about Thr210 phosphorylation is unclear. However, it most likely involves a transient ternary complex, as observed by cross-linking mass spectrometry (Lössl et al, 2016). The mechanism does not require canonical activation of Aurora-A on Thr288, but does require Bora that is phosphorylated on Ser112, which can act in *trans* to mimic activation loop phosphorylation (Fig. 1A) (Tavernier et al, 2021). It is not known how the Aurora-A/Bora complex interacts with PLK1 to catalyse its phosphorylation on Thr210.

High-resolution, experimental structural studies on transient interactions such as those involving PLK1 and Aurora-A remain challenging. We therefore took advantage of recent advancements in computational modelling that enable accurate predictions of the structures of proteins and their complexes (Bryant et al, 2022; Baek et al, 2021), and specifically AlphaFold3 which has the capability to include phosphorylated side-chains, ligands and ions (Abramson et al, 2024). We report a high-confidence structural model for the complex between PLK1, Aurora-A and Bora. The model was validated using structure-guided sequence variation, biochemical assays and NMR spectroscopy. It provides a rationale for the roles of Bora phosphorylation in the interaction, including an additional site modified by Aurora-A that we have characterised.

## Results

### Bora is a bridge between Aurora-A and PLK1 in the ternary complex

The complex of the three full-length human proteins (Bora (1–559), Aurora-A (1–403), PLK1 (1–603)) was modelled using AlphaFold2 (Mirdita et al, 2022) and AlphaFold3 (Abramson et al,

2024) (Fig. EV1A, AlphaFold3 model with Bora shown in magenta, Aurora-A in green and PLK1 in blue). The model had an overall interface predicted template modelling (ipTM) score of 0.63–0.65 using both AlphaFold3 and AlphaFold2 (Appendix Table S1). The ipTM score is a measure of the accuracy of the predicted relative positions of the residues in a complex, on a scale of 0–1. Bora was modelled as highly disordered, with residues 20–113 wrapped around the Aurora-A kinase domain. Residues 52–73 of Bora are modelled between the PLK1 kinase domain and Aurora-A kinase domain, with residues 58 to 68 predicted to form an alpha-helix. Unsurprisingly for an intrinsically disordered protein, the confidence of most of the Bora structural prediction was low, particularly over residues 175–559 (Fig. EV1B).

The model was simplified by removal of the low confidence regions of all three proteins, whilst preserving the key interactions. Bora 18–120 was wrapped around Aurora-A, forming the ternary interface with PLK1, and a short region of Bora (245–257) that includes the phosphorylated Ser252 site which interacted with the PBD of PLK1 (Fig. EV1C) (Chan et al, 2008). The PBD is located on the opposite side of the PLK1 kinase domain to the Bora interface. We therefore focussed on the kinase domains of both PLK1 (30–330) and Aurora-A (122–390), and the minimal region of Bora (18–120) required to stimulate Aurora-A (Tavernier et al, 2021). Models based on these truncated sequences perfectly conserved the interactions observed in the full-length protein model with Bora forming the core of the ternary interface (Fig. 1B, Bora in magenta between Aurora-A in green and PLK1 in blue). The average ipTM generated from 50 models produced with AlphaFold3 was 0.74 (Fig. 1C, plotted in green). Analysis of the ternary complex with AlphaBridge (preprint: Álvarez-Salmoral et al, 2024) identified two interfaces: between Aurora-A (130-280) and Bora (21–110), and between Bora (56–66) and PLK1 (86–103) (Fig. 1D). There is no direct interface between Aurora-A and PLK1, and so Bora can be considered as a bridge between the two kinases. Models generated using both AlphaFold2 and AlphaFold3 of the ternary complex were consistent (Appendix Fig. S1A,B).

Modelling the binary complexes produced consistent, high confidence models of Aurora-A bound to Bora but more variable, lower confidence models of PLK1 bound to either Bora or Aurora-A (Fig. 1C; Appendix Table S1). A similar trend was observed in the pairwise ipTM scores for the interfaces in the ternary complexes (Fig. 1E,F). All models of PLK1/Aurora-A complexes were of very low confidence.

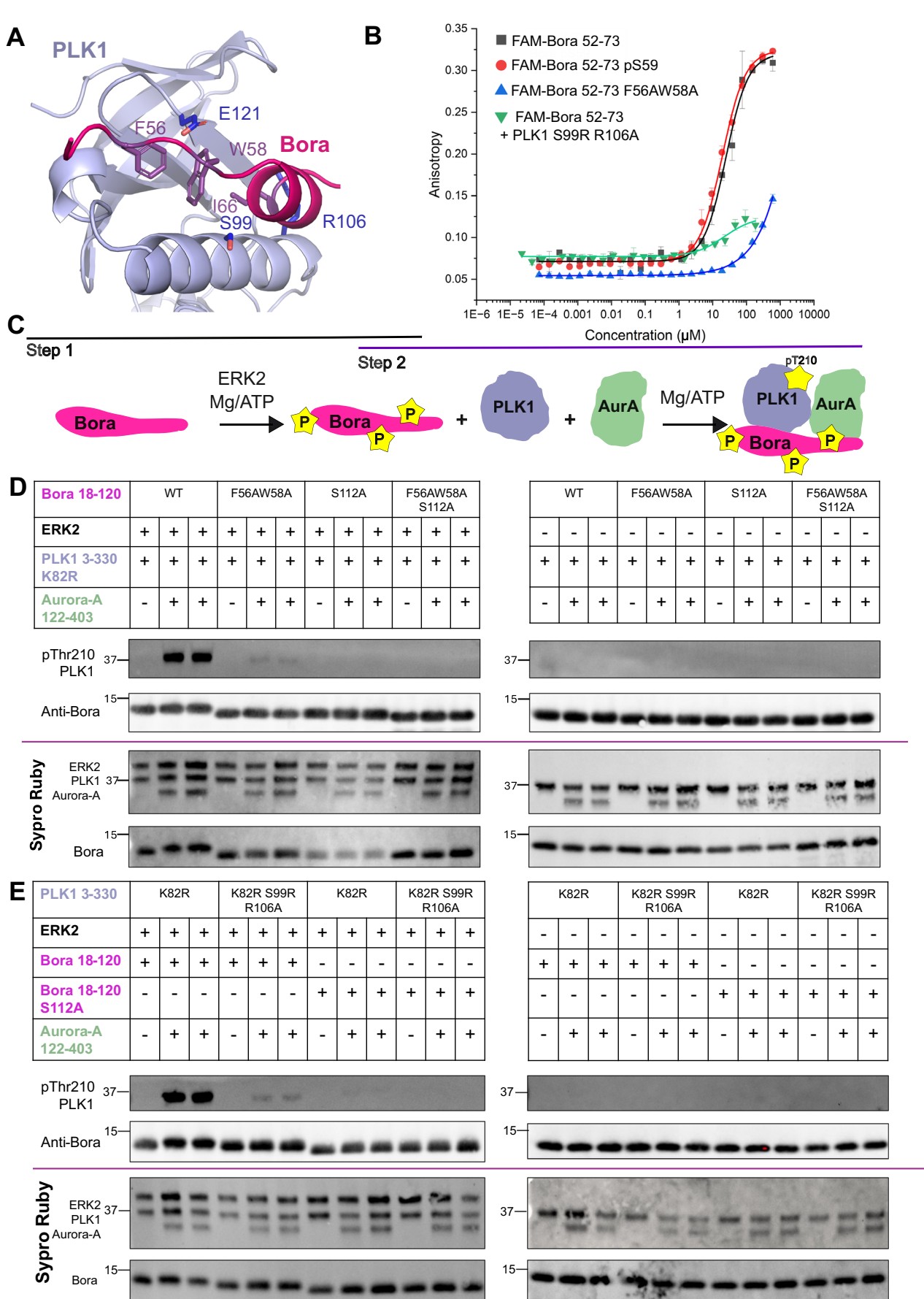

◄ **Figure 2.  Biochemical validation of the Bora:PLK1 interface.**

(A) Cartoon representation of the predicted interface between PLK1 kinase domain (blue) and Bora 18–120 (magenta). (B) Fluorescence anisotropy based direct binding assay looking at the interaction of FAM Bora 53–72 with PLK1 3–330 (K82R kinase dead version, shown in black, $K_d$ of 24 ± 3 μM), FAM Bora 53–72 pS59 (K82R kinase dead version, shown in red, $K_d$ of 19 ± 4 μM), FAM-Bora 53–72 F56A W58A (K82R kinase dead version, shown in blue) and FAM-Bora 53–72 with PLK1 mutated in the predicted interaction interface (K82R S99R R106A, shown in green). Data shown is the mean of three technical repeats, with the standard deviation shown as error bars. $n = 3$. (C) Schematic of the PLK1 phosphorylation assay, where Bora 18–120 is preincubated with ERK2 and ATP to produce Bora phosphorylated before Aurora-A and PLK1 are added in the second step. (D) Western blot analysis of the levels of phosphorylation of PLK1 kinase domain K82R at Thr210 when different Bora mutants are preincubated with ERK2 and added to wild-type dephosphorylated Aurora-A kinase domain (122–403). Inputs are shown with Sypro Ruby staining of nitrocellulose membranes. (E) Western blot analysis of the levels of phosphorylation of PLK1 kinase domain K82R and the S99R R106A mutant at Thr210 when Bora wild-type or Bora S112A is preincubated with ERK2 and added to wild-type dephosphorylated Aurora-A kinase domain. Inputs are shown with Sypro Ruby staining of nitrocellulose membranes. Source data are available online for this figure.

Bora wraps around the N-lobe of Aurora-A in all the models of the Aurora-A/Bora binary complex (AlphaBridge interface summary in Fig. EV1D and superposed models in Appendix Fig. S1C). In contrast, without Bora being 'tethered' in place by Aurora-A, the models of a complex between the PLK1 kinase domain and Bora 18–120 are highly variable (Appendix Fig. S2A,B). When Bora is limited to just residues (52–72) that are predicted to bind to PLK1, these models are more consistent, placing the Bora sequence at the same site in 9 out of 10 models (Appendix Fig. S2C, Phe56 and Trp58 shown in magenta).

The interface on PLK1 predicted to interact with Bora is analogous to the Y-pocket on Aurora-A, which interacts with Tyr8 and Tyr10 in TPX2 (Bayliss et al, 2003; McIntyre et al, 2017). Since this pocket in PLK1 is predicted to interact with a phenylalanine and tyrosine in Bora, this site will be provisionally labelled as the 'FW pocket'. Previous mass spectrometry analysis identified peptides within this region as evidence of an interaction between human PLK1 and human Bora, although their direct involvement in the interface was not shown (Bora peptides 50–78, 58–78 (Seki et al, 2008a)).

## Biochemical validation of the ternary complex model

Our model suggests that the interaction between Bora and PLK1 is mediated by a small section of Bora that interacts with the 'FW' pocket in PLK1 (Fig. 2A). A fluorescently labelled version of this short region of Bora (52–73) showed weak binding to 'kinase-dead' K82R PLK1 kinase domain in a fluorescence polarisation assay (Fig. 2B, shown in black), with a $K_d$ of 24 ± 3 μM. The interaction was substantially reduced in a variant of PLK1 with changes to the 'FW' pocket (R106A, S99R) (Fig. 2B, shown in green). Only weak binding was observed when a F56A W58A version of the Bora peptide was tested in the direct binding assay (Fig. 2B, shown in blue).

The model was validated further using an in vitro assay with phosphorylation of Thr210 of PLK1 as a readout. Mutations were introduced into Bora 18–120 to remove two of the hydrophobic residues that are predicted to interact with PLK1 (Phe56 and Trp58, Fig. 2A). Purified mutated Bora was pre-phosphorylated with ERK2, before the addition of wild-type unphosphorylated Aurora-A kinase domain and PLK1 3–330 K82R as the substrate (Fig. 2C). ERK2 kinase was chosen as it is selective for (S/T)P motifs equivalent to those phosphorylated in vivo by CDK1 but can phosphorylate this shorter sequence of Bora which lacks the Cy-motif needed to recruit the CDK1–Cyclin complex (Tavernier et al, 2021). When WT Bora was pre-phosphorylated by ERK2, phosphorylation of PLK1 at Thr210 was stimulated by Aurora-A,

consistent with previous results (Fig. 2D, lanes 2 and 3) (Tavernier et al, 2021). There was a significant reduction in PLK1 phosphorylation at Thr210 when the pre-phosphorylated F56A W58A mutant of Bora was used (Fig. 2D, lanes 5 and 6, quantified in Fig. EV2A). A similar effect was seen when Ser112 in Bora was mutated to alanine (Fig. 2D, lanes 8 and 9) and when the mutations were combined (Fig. 2D, lanes 11 and 12). A Bora variant in which only S112 can be phosphorylated by ERK2 (Bora S27A S41A T52A, labelled as ERK mutant) also led to clear stimulation of Aurora-A activity, that was abrogated upon inclusion of the F56A W58A mutation (Appendix Fig. S3A). Unexpectedly, the F56A W58A Bora was less efficiently phosphorylated on S112 by ERK2 than the WT Bora (Appendix Fig. S10B compared to J and Appendix Table S2), and this was also apparent in the ERK mutant variants (Appendix Fig. S10F compared to H and Appendix Table S2). To mitigate for this, the assay used a 5-fold molar excess of Bora compared to levels of PLK1 and Aurora-A to ensure that enough phosphorylated Bora was present.

There was a significant reduction in levels of phosphorylation on PLK1 at Thr210 when the 'FW' pocket mutant of PLK1 (S99R R106A) was used as a substrate (Fig. 2E, comparing lanes 2 and 3 with lanes 5 and 6, quantified in Fig. EV2B). These mutations did not impact the overall structure of PLK1, as the 'FW' variant retained interaction with a DARPin that binds to the kinase domain (Appendix Fig. S3B,C) (Bandeiras et al, 2008).

We conclude that the interaction of the region of Bora centred on Phe56/Trp58 is critical for its interaction with PLK1, and for the subsequent phosphorylation of Thr210 by Aurora-A.

## Conservation of the Bora–PLK1 interaction site

Examination of the predicted interface between PLK1 and Bora revealed a pattern of conserved residues (Fig. 3A–D). Orthologues of the human sequences were identified using a PSI-Blast search and the sequences aligned using MAFFT (Appendix Figs. S4 and 5). The residues that are predicted to interact with Bora in PLK1 were selected using a PDBe PISA (Krissinel and Henrick, 2007) analysis of the model of the ternary complex. Lys86, Leu89 and Arg95 at the interface are well conserved in PLK1 (human numbering, Fig. 3B). Furthermore, the two hydrophobic residues that point into the FW pocket (Phe56 and Trp58) are conserved between all Bora orthologues (Fig. 3D), as is the proline at the end of the short helix in Bora in the complex (Pro68). This suggests that the interaction between Bora and PLK1 using this interface is likely to be similar in many organisms. To assess this hypothesis, AlphaFold3 was used to model the orthologues from an organism distant in evolution from humans, *Strongylocentrous*

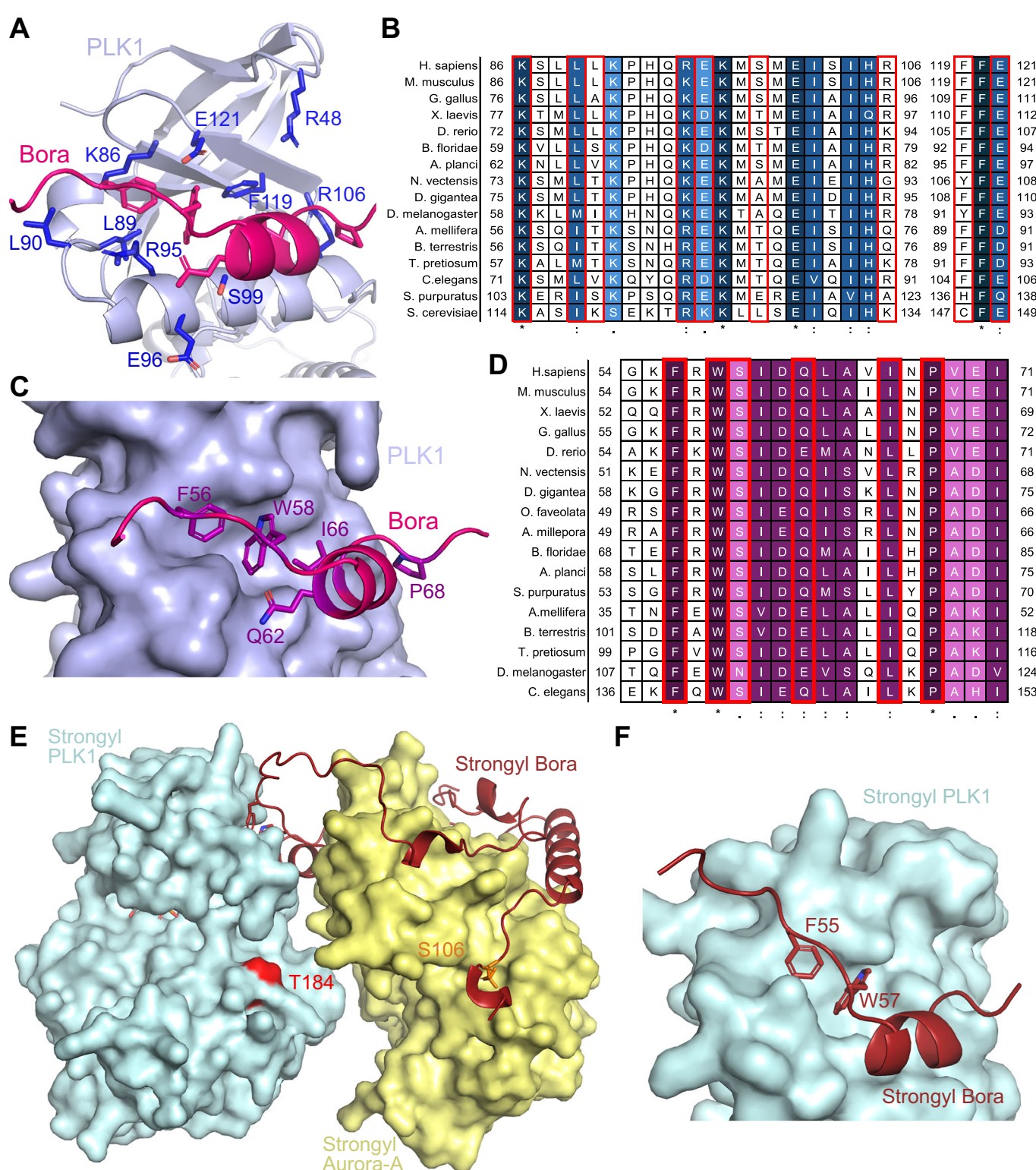

*purparatus* (sea urchin). The truncated ternary complex has a very similar arrangement to the human ternary complex (Fig. 3E compared to Fig. 1B). The interface between PLK1 and Bora is conserved, with Phe55 and Trp57 from Bora pointing into the surface of PLK1 (Fig. 3F, sea urchin).

## Characterisation of Bora 1–120 and its interaction with Aurora-A using NMR

$^{15}$N-$^{13}$C labelled Bora 1–120 was expressed and purified prior to characterisation by NMR. The $^{1}$H-$^{15}$N HSQC spectrum shows features of

Figure 3. Conservation of the Bora:PLK1 interface.

(A) Model of Bora (magenta) interacting with the FW pocket on PLK1 (blue), with residues in the interface from the PLK1 side coloured based on conservation. The darker the residue, the more conserved it is. (B) Selected regions of the MAFFT sequence alignment of the PLK1 kinase domain from diverse species. The residues highlighted in red are at the predicted interface with Bora. (C) Model of Bora (magenta) interacting with the FW pocket on PLK1 (blue), with residues in the interface from the Bora side coloured based on conservation. The darker the residue, the more conserved it is. (D) Sequence alignment of the region of Bora that is predicted to interact with PLK1. The darker the residue, the more conserved it is. The residues highlighted in red are at the predicted interface with PLK1. (E) AlphaFold3 model of the ternary complex from *Strongylocentrotus purpuratus* with PLK1 shown in cyan, Bora shown in dark red and Aurora-A in yellow. Thr184 that is predicted to be phosphorylated by Aurora-A is shown in bright red on the surface of PLK1, with the phosphorylated serine in Bora shown in orange. (F) View of the interface between PLK1 (cyan) and Bora (red) in *Strongylocentrotus purpuratus* orthologues, highlighting the conserved Phe55 and Trp57 in the pocket on the surface of PLK1.

a disordered protein as seen previously (Tavernier et al, 2021), albeit there is enough peak dispersion to suggest the sequence may contain some elements of order (Fig. 4A, spectra in black). The $^1$H-$^{15}$N HSQC was straightforwardly assigned using triple resonance experiments (Appendix Fig. S6). Two regions in the C-terminal half of Bora 1–120 (Pro73-Arg87 and Lys90-Thr105) have significant helical propensity (regions exhibit positive Cα and CO secondary shifts and negative Cβ secondary shifts, Fig. EV3A–D). Regions with weaker helical propensities are also observed in the N-terminal half (Tyr31-Thr38 and Ile60-Val65). The helical propensity regions in the unbound Bora NMR data match up very well with the helical regions of Bora present in the Aurora-A/Bora and Aurora-A/Bora/PLK1 AlphaFold models. This finding supports the AlphaFold models and indicates that the interaction builds upon latent structure within the 'disordered' Bora chain.

Residues in the two high-helical-propensity regions stand out from the rest of the Bora sequence by having elevated $^{15}$N $R_2$ relaxation rates (and thus, comparatively low peak intensities at 10 °C, Fig. EV3E,G) and elevated hetNOE $^{15}$N-$^1$H values (Fig. EV3F). These two regions also maintain or increase their $^1$H-$^{15}$N HSQC peak intensities on increasing temperature, whereas intensities for residues elsewhere within Bora 1–120 tend to reduce with temperature (Fig. EV3H). The region of Bora predicted to interact with both Aurora-A and PLK1 (close to Ile60) has similarly elevated relaxation features. The hetNOE features in the C-terminus indicate deviation away from the more freely dynamic, disordered behaviour seen in the N-terminal half of Bora 1–120, to areas with restricted motion on the ps–ns timescale from partial helix formation. The elevated $R_2$ values could also contain a contribution from some slower ms–ms dynamics in these regions.

Interaction with a binding partner leads to changes in the chemical environment and relaxation/dynamic behaviour, resulting in NMR spectral changes in peak position and intensity, respectively; the largest changes are typically observed at the binding site(s). When Aurora-A 122–403 (kinase domain) was titrated into $^{15}$N-$^{13}$C labelled Bora 1–120, no clear chemical shift perturbations were observed, but loss in peak intensities, to the point of disappearances, were seen at positions all through the sequence (Fig. 4A, shown in red). Peak intensity changes seen across the Bora sequence at a [Bora]:[AurA] molar ratio of 2:1 are shown in Fig. 4B. The Bora residues with the most significant loss in peak intensity, which are likely to be those most constrained by the interaction, match very well with those predicted to most closely interact (Fig. 4C, blue).

## Phosphorylation at Ser112 stabilises the interaction between Aurora-A and Bora

AlphaFold3 supports the modelling of post-translational modifications (PTMs) such as phosphorylation. When the truncated ternary complex between Bora, Aurora-A and PLK1 was modelled with phosphorylation of Bora at Ser112, a site which has been shown to act in *trans* to activate Aurora-A (Tavernier et al, 2021), the ipTM score was slightly improved (Fig. 1C, shown in dark blue, average ipTM 0.76 compared to 0.74). The phosphoryl group at this site is predicted to interact with Arg180, Arg286 and Thr288 in Aurora-A (Fig. 4D). This is comparable to how phosphorylated Thr288 in the activation loop of Aurora-A is part of a stabilised activation loop when TPX2 is bound (Bayliss et al, 2003).

The effect of pSer112 in the pairwise ipTM scores were also analysed using 50 models produced with AlphaFold3 (5 models from each of 10 runs). There is on average a 0.05 uplift in ipTM score when looking at the model of Aurora-A bound to pSer112 (Fig. 1E, shown in blue). Whereas the presence of the phosphorylation of on Bora pS112 doesn't affect the pairwise Bora-PLK1 ipTM score significantly (Fig. 1F, shown in blue). This is consistent with a specific effect of pSer112 on stabilisation of the interaction with Aurora-A.

To further investigate the role of this phosphorylation in vitro, we assigned and then phosphorylated $^{15}$N-$^{13}$C labelled Bora 18–120 S27A S41A T52A (ERK mutant) with the kinase ERK. In $^1$H-$^{15}$N HSQC spectra, ERK phosphorylation of Bora 18–120 S27A S41A T52A specifically targets Ser112 (Fig. EV4A). When Aurora-A kinase domain is titrated into this phosphorylated Bora, we see a similar profile but with larger intensity losses in the pSer112 region of Bora, suggesting a stronger local interaction with Aurora-A (Fig. 4E). As well as a decrease in intensity around pSer112 in Bora, we see an overall effect with decreased intensity across most of the Bora sequence. The increased binding at pSer112 potentially acts as a tether holding Aurora-A and Bora together, improving the stability of the interaction overall. This model is in agreement with the observed higher affinity of Aurora-A for phosphorylated Bora (Tavernier et al, 2021). We also looked at the Bora 18–120 F56A W58A mutant with NMR. From Cα secondary shifts the F56A W58A mutant and the ERK mutant have very similar structural propensities to WT indicating that the effect of mutations is only local. By partially phosphorylating the F56A W58A mutant with ERK2 to give a roughly 1:1 mixture of Bora F56A W58A Ser112 and pSer112 and then quenching kinase activity, we could subsequently compare the Aurora-A binding ability of the two phosphorylation states in one experiment (Appendix Fig. S7). While most peaks for the two species overlay completely, there are distinct peaks for pSer112/Ser112 and a handful of neighbouring residues. More significant peak intensity losses were observed for all the peaks associated with the pSer112 state of Bora F56A W58A when compared with the unphosphorylated Ser112 state. This indicates a stronger interaction of Aurora-A with the Bora F56A W58A pSer112 state (mirroring the ERK mutant results). The result also

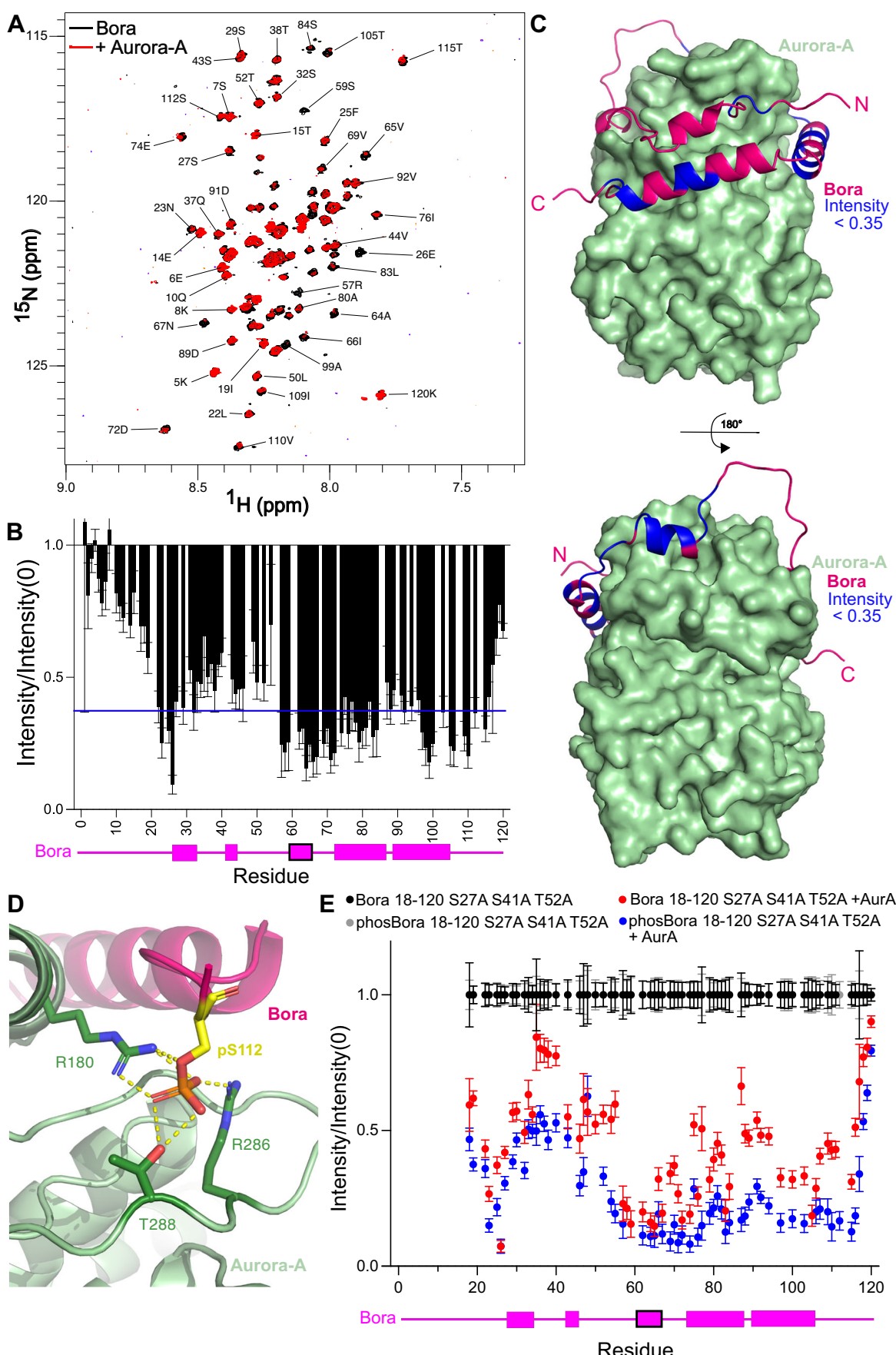

**Figure 4. NMR analysis of Bora interaction with Aurora-A.**

(**A**) $^1H$–$^{15}N$ HSQC spectra recorded of *Homo sapiens* Bora 1–120 in the absence (black) and presence of human Aurora-A 122–403 (red). A significant number of Bora peaks disappeared in the presence of Aurora-A. (**B**) The relative peak intensity changes for residues in Bora 1–120 upon addition of Aurora-A 122–403 ([Bora]:[AurA] = 2:1). Error bars ($\Delta R$) for peak intensity ratios ($R$) were determined from a measure of the baseline noise ($\Delta z$) in each spectrum, using $(\Delta R/R)^2 = (\Delta z_1/z_1)^2 + (\Delta z_2/z_2)^2$. Each spectrum was recorded from one sample ($n = 1$). The model-predicted secondary structure of Bora in the complex is represented below in pink, with predicted alpha helices shown as rectangles. The helix predicted to interact with the FW pocket of PLK1 is bordered in black. (**C**) Mapping the residues involved in binding onto the predicted structure of Bora 1–120 bound to Aurora-A. Aurora-A is shown in green, with Bora represented in magenta. Residues in Bora that show a significant loss in peak volume upon inclusion of Aurora-A are shown in blue (intensity change below 0.35). (**D**) Analysis of the AlphaFold3 model of Bora phosphorylated at Ser112 (yellow) bound to Aurora-A (shown in green). Phosphorylated Ser112 in Bora is predicted to co-ordinate two arginine residues in Aurora-A as well as Thr288 in the activation loop of Aurora-A. (**E**) Change in peak intensity in the $^1H$–$^{15}N$ HSQC spectrum of $^{15}N$-labelled Bora 18–120 S27A S51A T52A with and without phosphorylation of Bora by ERK at Ser112 (black—no phosphorylation, grey—with phosphorylation by ERK) and when Aurora-A is present (red—unphosphorylated Bora and Aurora-A, blue—phosphorylated Bora with Aurora-A). The reduction in peak intensities around Ser112 (and at other parts of the sequence) is higher for the phosphorylated version. Error bars ($\Delta R$) for peak intensity ratios ($R$) were determined from a measure of the baseline noise ($\Delta z$) in each spectrum, using $(\Delta R/R)^2 = (\Delta z_1/z_1)^2 + (\Delta z_2/z_2)^2$. Each spectrum was recorded from one sample ($n = 1$). The model-predicted secondary structure of Bora in the complex is represented below in pink, with predicted alpha helices shown as rectangles. The helix predicted to interact with the FW pocket of PLK1 is bordered in black. Source data are available online for this figure.

suggests that the reduced phosphorylation efficiency of ERK for the Bora F56A W58A mutant is not the limiting effect when it comes to PLK1 phosphorylation since binding the pSer112 state is favoured.

Crystal structures are available of Aurora-A in complex with several other binders: TPX2, CEP192, TACC3 and N-Myc (Bayliss et al, 2003; Richards et al, 2016; Holder et al, 2024; Burgess et al, 2018; Park et al, 2023). In the AlphaFold models Bora is predicted to exploit the same set of pockets on the N-lobe of Aurora-A as TPX2, CEP192 and TACC3 using similar interactions (Fig. 5A). In the overlaid structures, Bora Phe25 overlaps with Phe19 in TPX2 and Phe490 in CEP192 (Fig. 5B), and Bora Phe103/Phe104 overlaps with Trp34/Phe35 from TPX2 (Fig. 5E). Phe45 in Bora is predicted to interact with the Y-pocket in Aurora-A, although it does not closely resemble Tyr8 and Tyr10 of TPX2 at this site (Fig. 5C). Ile71 of Bora is predicted to bind in a pocket on the other side of the Aurora-A N-lobe that can be occupied by either Phe525 of TACC3 or Phe508 of CEP192 (Fig. 5D).

An NMR-based competition assay was used to probe the Aurora-A/Bora interaction sites in solution. $^1H$–$^{15}N$ HSQC spectra of $^{15}N$-$^{13}C$ labelled Bora 1–120 were again recorded before and after the addition of Aurora-A, resulting in a decrease in the intensities across the Bora sequence (Fig. 6A). Next, Aurora-A binding proteins with known binding sites were added (TPX2 1–43, CEP192 442–533) and spectra re-measured, resulting in the partial rescue of intensities (Fig. 6B,C). This indicates that TPX2 and CEP192 can compete for Aurora-A-binding with Bora – liberating the Bora leads to peak intensity recovery. The recovery on addition of TPX2 was most marked in the region around Bora Phe25, and CEP192 was more effective for peak recovery of Bora's C-terminal region. These observations are consistent with the predicted binding sites of Bora compared to those utilised by TACC3, CEP192 and TPX2 on Aurora-A (Fig. 6D,E).

## Aurora-A phosphorylation of Bora Ser59 enhances PLK1 activation

Given that Bora is heavily phosphorylated by numerous kinases, we considered whether modification of any site other than Ser112 could influence the interaction with PLK1 or Aurora-A. A peptide array covering the Bora sequence was used to identify sites that can be phosphorylated by Aurora-A (Fig. 7A). Bora peptides containing Ser59 exhibited the most significant increase in staining (Fig. 7A). This site has previously been identified, with 100% modification

observed after 20 min of Aurora-A incubation with Bora (Lössl et al, 2016). This residue is conserved in most species, but not *Drosophila melanogaster* (Fig. 3D), where an asparagine is present at this position. The consensus sequence for substrate phosphorylation by Aurora-A is R-X-S/T-B, with B denoting any hydrophobic residue with the exception of Pro (Alexander et al, 2011; Ferrari et al, 2005). This matches to the sequence in Bora around Ser59 (F-R-W-**S**-I).

The predicted effect of Bora phosphorylation at Ser59 to the interaction with Aurora-A and PLK1 in the predicted ternary complex was modelled using AlphaFold3. The model indicates that phosphorylation of Ser59 makes an additional interaction with Arg95 of PLK1 and Arg205 of Aurora-A (Fig. 7B). This may stabilise a ternary complex further to facilitate PLK1 phosphorylation. To ascertain whether phosphorylation of Ser59 enhances the interaction of Bora with PLK1, a phosphorylated version of the FAM-labelled Bora 52–73 peptide was produced. When this was tested in the direct binding assay with PLK1, the binding was very similar to that of the wild-type Bora sequence (Fig. 2B, shown in red. $K_d$ 19 ± 4 μM).

The phosphorylation of $^{15}N$-labelled human Bora 1–120 was monitored using NMR. Clean, specific phosphorylation at Ser59 was observed through a substantial downfield $^1H$ shift in the peak for this residue when Bora 1–120 was incubated with Aurora-A (Fig. 7C, full spectra in Fig. EV4B). Ser59 phosphorylation resulted in chemical shift perturbations (CSPs) for residues Thr52-Ile71, and an unusual upfield $^{15}N$ shift for Ser59. This may indicate an increase in local helical propensity in this region of Bora, consistent with the helical conformation predicted in the ternary complex model.

In a similar fashion to the pS112 phosphorylated Bora, when Aurora-A was titrated into pS59 phosphorylated Bora, larger intensity losses in the region of the phosphorylation site are observed when compared to unphosphorylated Bora (Fig. EV5A,B). This suggests a stronger local Bora–Aurora-A interaction after phosphorylation at Ser59.

To probe the functional relevance of Ser59, it was mutated to alanine and the variant protein used in an in vitro assay of PLK1 phosphorylation at T210 (Fig. 7D). Following incubation with ERK-phosphorylated Bora S59A and wild-type Aurora-A 122–403, PLK1 phosphorylation after 30 min showed a 50% reduction compared to using wild-type Bora (Fig. 7E,F). This indicates that Bora Ser59 phosphorylation mediated by Aurora-A increases the

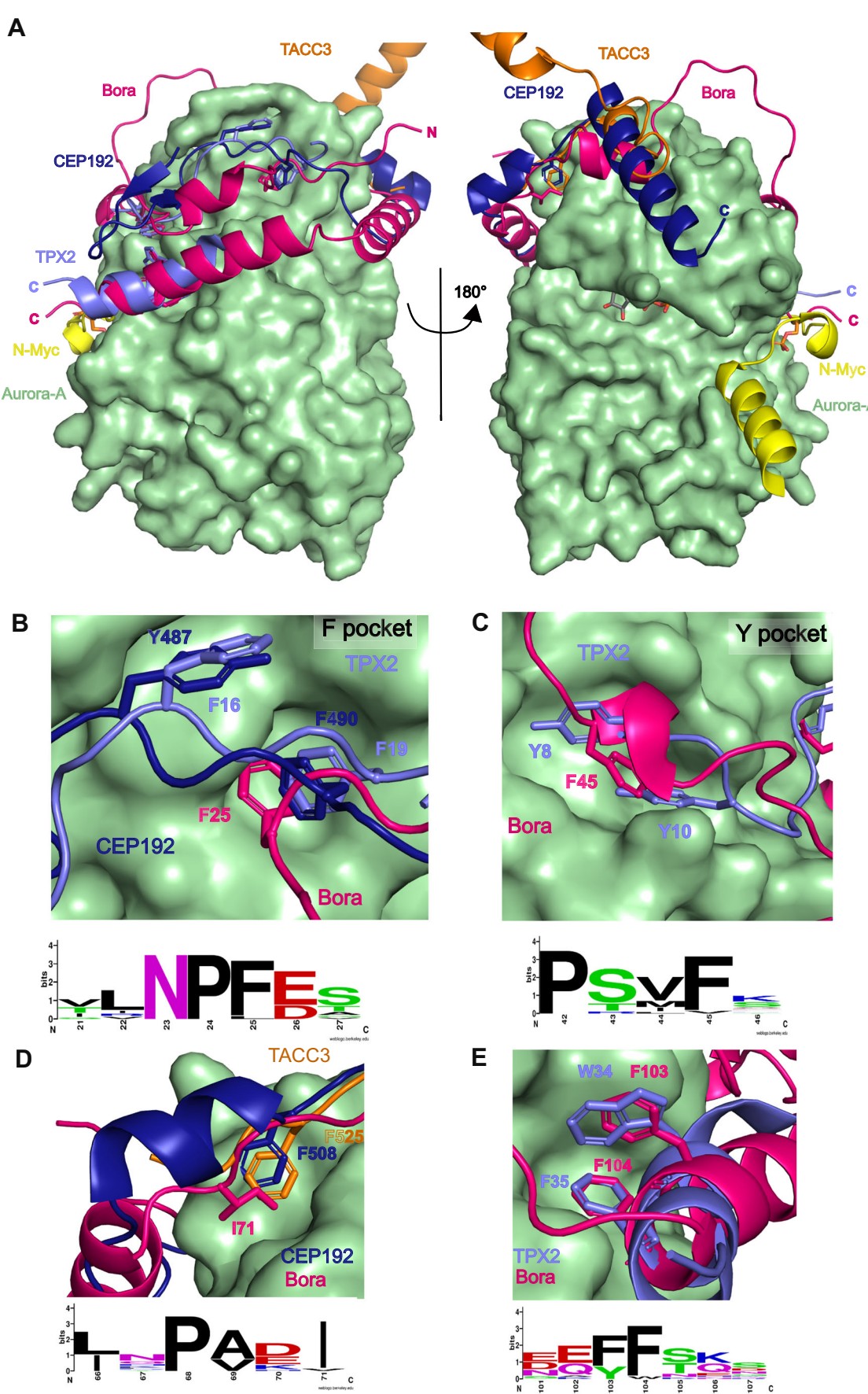

**Figure 5. Structural comparison of Aurora-A complexes.**

(A) Comparison of the model of Bora (magenta) bound to Aurora-A (green) with the crystal structures of Aurora-A complexes. TPX2 1–43 is shown in light blue (PDB 1OL5), CEP192 468–533 in dark blue (PDB 8PR7), N-Myc 61–89 in yellow (PDB 5G1X) and TACC3 in orange (PDB 5ODT). (B) Comparisons of predicted interaction of Bora with Aurora-A, focusing on the F-pocket of Aurora-A. Phe25 of Bora is predicted to bind in the F-pocket on the surface on Aurora-A, which can also be occupied by Phe19 in TPX2 or Phe490 in CEP192. The conservation of these sites in Bora is shown as a WebLogo below. (C) Comparison of the predicted interaction of Bora with Aurora-A in the Aurora-A 'Y-pocket'. Bora Phe45 is predicted to bind to Aurora-A at this site, utilising the pocket that is occupied by Tyr8 and Tyr10 in TPX2. The conservation of these sites in Bora is shown as a WebLogo below. Phe45 in Bora is highly conserved. (D) Comparison of the predicted interaction of Bora with Aurora-A, focusing on the pocket at the top of Aurora-A that both TACC3 and CEP192 interact with. TACC3 and CEP192 have Phe residues that interact with a pocket on the top of the N-lobe of Aurora-A (Phe525 and Phe508, respectively). The modelling predicts that Ile71 from Bora occupies this site on Aurora-A. The conservation of these sites in Bora is shown as a WebLogo below. (E) Comparison of the predicted interaction of Bora with Aurora-A, focusing on the region near the activation loop. Phe103 and Phe104 of Bora are predicted to interact with Aurora-A similarly to how TPX2 Trp34 and Phe35 interact with the surface of Aurora-A. The conservation of these sites in Bora is shown as a WebLogo below.

efficiency of PLK1 phosphorylation. Given that this residue is not present in *Drosophila melanogaster*, we wondered whether other residues at the interface could help stabilise the *Drosophila* ternary complex. We modelled the ternary complex on AlphaFold3 and compared this to the model of the human complex, which was overlaid on the PLK1 kinase domain (Appendix Fig. S8A). At the interface between *Drosophila* Bora and PLK1, an extra salt bridge is predicted between Lys120 in Bora and Glu93 in PLK1, whereas in the human sequence this wouldn't be able to form as Lys120 is replaced with a Asn (Appendix Fig. S8B). Thus, it appears that the *Drosophila* complex has an additional, constitutive salt-bridge that might compensate for the absence of the phospho-Ser59-dependent salt-bridge present in most species.

## Discussion

It has proven challenging to determine experimental structures of kinase–substrate complexes, and only thirty have been deposited in the PDB, eleven of which are kinase autophosphorylation structures (preprint: Faezov and Dunbrack, 2023). The mechanism by which Aurora-A recognises PLK1 via Bora has also eluded experimental structure determination, due to disfavourable properties of the system: Bora is a disordered protein that requires phosphorylation on specific sites to interact with and activate Aurora-A, and the PLK1/Bora interaction is transient, like most kinase–substrate pairs (Bruinsma et al, 2015; Tavernier et al, 2021).

Here we have used AlphaFold3 to model how Bora, Aurora-A and PLK1 come together to phosphorylate PLK1 on Thr210. Bora is predicted to form a bridge to bring the Aurora-A and PLK1 together, through interactions with pockets on the surfaces of the N-lobes of the kinases. A critical interaction is formed between a motif in Bora (56–66) and a pocket at the C-helix of PLK1. Phosphorylation of Bora on Ser112 is important for the phosphorylation of PLK1 as it mimics the structural role of Aurora-A activation loop phosphorylation in the context of a TPX2-like binding motif and in the context of unphosphorylated Aurora-A. Aurora-A phosphorylation of Bora Ser59 also enhances the efficiency of PLK1 phosphorylation. Ser59 is a good substrate for Aurora-A, is highly conserved, and lies within the critical motif of Bora positioned at the interface with PLK1 in the ternary complex. The predicted interface between Bora and the PLK1 kinase domain is small (buried surface area of $574 \pm 39\ \text{Å}^2$ with unphosphorylated Bora, $598 \pm 42\ \text{Å}^2$ with pS59 pS112 Bora), consistent with a weak interaction. In contrast, the predicted

interface between Aurora-A and Bora is large (buried surface area of $2739 \pm 181\ \text{Å}^2$ with unphosphorylated Bora and $2663 \pm 126\ \text{Å}^2$ with pS59 pS112 Bora).

One limitation of the modelling is that, although the activation loop of PLK1 faces the active site of Aurora-A, Thr210 is too distant for a productive phospho-transfer reaction. There is some flexibility in the PLK1 position, indicating that movement of PLK1 Thr210 towards the active site is possible, but it would not be sufficient to close the gap. This is because the activation loop of PLK1 adopts a closed, active-like conformation, not an open/extended conformation that would be needed to act as a substrate. Attempts to model PLK1 with a more extended activation loop conformation, using a range of templates, were unsuccessful. AlphaFold3 is ill-suited for predicting dynamic features of the complex, as it has been trained against stable, experimentally determined structures in the PDB, in which transient conformations of kinase–substrate complexes are underrepresented.

This research is supported by the timely release of two parallel publications which also study the mechanism by which PLK1 phosphorylation is simulated by Bora in complex with Aurora-A (Esposito-Verza et al, 2026; Pillan et al, 2026). These two works complement ours, also aiming to validate a ternary complex modelled with AlphaFold, albeit using differing approaches to do so, such as a novel co-expression system entitled MITOKINAC to screen mutations within the predicted complex (Pillan et al, 2026). Esposito-Verza et al also highlight the lack of an optimal consensus sequence in PLK1, potentially explaining the unique requirement for Bora to facilitate this phosphorylation.

The Aurora-A/Bora interaction was previously proposed to resemble that of the TPX2/Aurora-A interaction, based on the sequence similarities of two motifs. Motif one, comprising 22–35, was proposed to resemble TPX2 7–20, but whereas TPX2 binds the Y-pocket via Tyr8 and Tyr10, the sequence alignment of Bora with TPX2 placed only Phe25 of Bora into the Y-pocket of Aurora-A, preceded by an "NP" sequence that does not resemble the "YS" of TPX2. In the AlphaFold3 model of the ternary complex, the chain direction of Bora is opposite to that of TPX2, and Phe25 of Bora is predicted to interact with the F-pocket in Aurora-A. Motif two in the previous study, comprising Bora 100–111, was predicted to bind to Aurora-A like TPX2 31–42. This hypothesis is supported by the AlphaFold3 model, in which Phe103 and Phe104 of Bora are structurally equivalent to Trp34 and Phe35 of TPX2. Both TPX2 and Bora appear to stabilise the activation loop of Aurora-A, either by facilitating interactions between a phosphorylated residue in the activation loop (Thr288) and the rest of Aurora-A, or by acting in

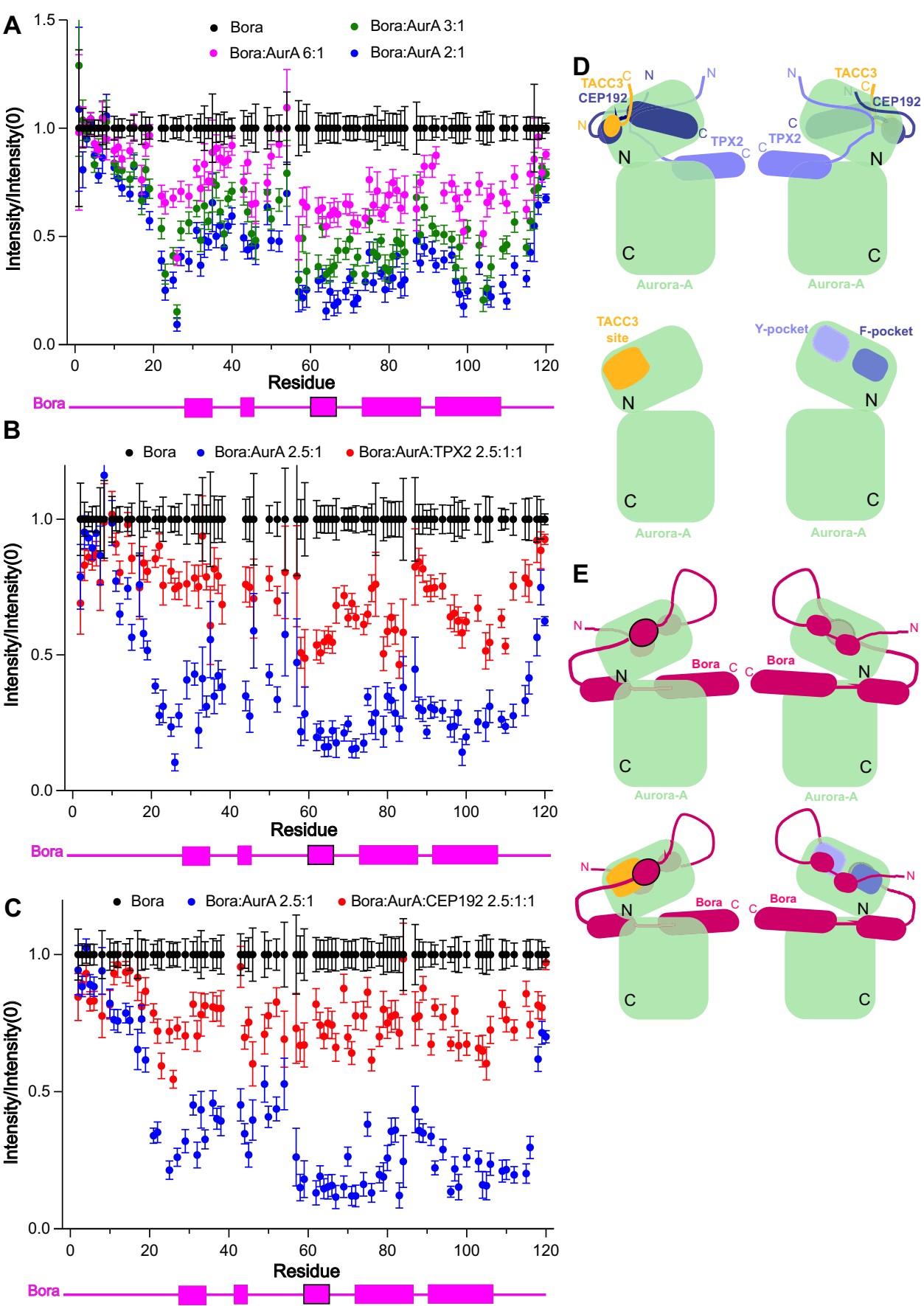

**Figure 6. NMR-based competition assays for binding sites on Aurora-A.**

(A) The relative change in intensity for peaks assigned to residues in Bora 1–120 in the presence of increasing amounts of Aurora-A kinase domain. Aurora-A interaction led to peak intensity loss at sites of interaction but no clear chemical shift perturbations. Error bars ($\Delta R$) for peak intensity ratios ($R$) were determined from a measure of the baseline noise ($\Delta z$) in each spectrum, using $(\Delta R/R)^2 = (\Delta z_1/z_1)^2 + (\Delta z_2/z_2)^2$. Each spectrum was recorded from one sample ($n = 1$). The model-predicted secondary structure of Bora in the complex is represented below in pink, with predicted alpha helices shown as rectangles. The helix predicted to interact with the FW pocket of PLK1 is bordered in black. (B) Aurora-A competition assay of Bora 1–120 with TPX2 1–43. Labelled Bora 1–120 is incubated with Aurora-A (2.5:1 molar ratio), leading to position-dependent peak intensity loss as the Bora:Aurora-A interactions occur (as per (A) shown in blue). Introduction of unlabelled TPX2 1–43 (1:1 molar ratio with Aurora-A) then leads to rescue of the Bora signal. Error bars ($\Delta R$) for peak intensity ratios ($R$) were determined from a measure of the baseline noise ($\Delta z$) in each spectrum, using $(\Delta R/R)^2 = (\Delta z_1/z_1)^2 + (\Delta z_2/z_2)^2$. Each spectrum was recorded from one sample ($n = 1$). (C) Aurora-A competition assay of Bora 1–120 with CEP192 468–533. Labelled Bora 1–120 is incubated with Aurora-A (2.5:1 molar ratio), leading to position-dependent peak intensity loss as Bora:Aurora-A interactions occur (as per (A) shown in blue). Introduction of unlabelled CEP192 (1:1 molar ratio with Aurora-A) then leads to rescue of the Bora signal. Error bars ($\Delta R$) for peak intensity ratios ($R$) were determined from a measure of the baseline noise ($\Delta z$) in each spectrum, using $(\Delta R/R)^2 = (\Delta z_1/z_1)^2 + (\Delta z_2/z_2)^2$. Each spectrum was recorded from one sample ($n = 1$). (D) Illustration of the sites used by TACC3 (orange), TPX2 (light blue) and CEP192 (dark blue) to interact with Aurora-A (light green). The TACC3 binding site, Y-pocket and F-pocket are highlighted on the Aurora-A N-lobe. (E) Illustration of the predicted sites used by Bora (magenta) to bind to Aurora-A (light green). The TACC3 binding site, Y-pocket and F-pocket are highlighted on the Aurora-A N-lobe. The helix predicted to interact with the PLK1 'FW' pocket is bordered in black in the images on the left. Source data are available online for this figure.

*trans* and providing a phosphate to support this stabilisation (pSer112 in Bora).

Crystal structures of the chromosome passenger complex protein INCENP bound to the Aurora-B and C kinase domains show that it too wraps around the N-lobe, and phosphorylation of Ser893 and Ser894 in the INCENP sequence is vital for the complete stimulation of kinase activity (Abdul Azeez et al, 2019; Elkins et al, 2012). The structure of INCENP bound to Aurora-C and an ATP-competitive inhibitor BRD-7880 (PDB 6GR8) was aligned with the model of Aurora-A bound to Bora, highlighting three points of similarity in these interfaces (Appendix Fig. S9). Short helices in INCENP and Bora bind above the kinase C-helix via similar hydrophobic interactions, namely Phe45 of Bora compared with Phe881 in INCENP (Appendix Fig. S9B). Another similarity is the helix that is predicted to wrap around the N-lobe (Appendix Fig. S9C). Bora Ile76 and Gln79 overlay on top of INCENP Ile855 and Gln585, respectively, suggesting Bora has a very similar mode of binding to this region of the kinase domain of Aurora proteins. Finally, near the activation loop, the phosphorylated Ser893 and Ser894 in INCENP are positioned at a similar site to pSer112 in Bora, with Ser893 and Ser894 interacting with the activation loop of Aurora-C (Appendix Fig. S9D). Thus, the way that Bora interacts with Aurora-A combines features similar to both INCENP and TPX2.

A recent publication has also highlighted the importance of Bora phosphorylation on Ser59 for the successful initiation of mitosis (Zhu et al, 2025), albeit the authors suggest that PKA is responsible for producing this pool of phosphorylated Bora. It is possible that both Aurora-A and PKA mechanisms for Bora phosphorylation can co-exist, providing more than one way to enhance the interaction between Aurora-A and Bora. Interestingly, we have observed with NMR that unphosphorylated Aurora-A T288V is able to phosphorylate pSer112 Bora on Ser59, consistent with a model of Aurora-A/Bora/PLK1 complex assembly and function prior to, and without the need for, Aurora-A phosphorylation on Thr288.

The identification of selective kinase inhibitors remains a significant challenge (Karaman et al, 2008). We have shown that the FW pocket on PLK1 mediates its interaction with Bora. This pocket is structurally analogous to the Y-pocket of Aurora-A and the PDK1-interacting fragment (PIF) pocket in PDK1—both known regulatory sites that influence kinase activation and selectivity (Biondi et al, 2000; Balendran et al, 1999). Ongoing efforts to develop selective binders for these pockets (Rettenmaier et al, 2014; Engel, 2013; Arencibia et al, 2017; Stockwell et al, 2024; McIntyre et al, 2017) suggest that targeting the FW pocket could similarly disrupt PLK1 activation, offering a potential route to a new class of specific PLK1 inhibitors for PLK1-dependent cancers.

## Methods

### Reagents and tools table

| Reagent/Resource | Reference or Source | Identifier or Catalog Number |
| --- | --- | --- |
| **Experimental models** | | |
| B834 competent *E. coli* cells | Novagen | |
| DH5α competent *E. coli* cells | NEB | |
| BL21(DE3) RIL competent *E. coli* cells | Novagen | |
| **Recombinant DNA** (pet30TEV vector has TEV cleavable 6xHistag) | | |
| pCDF Lambda phosphatase | This study | |
| pet30TEV human Aurora-A 122–403 C290A C393A | Burgess and Bayliss (2015) | |
| pet30TEV human Aurora-A 122–403 | This study | |
| pGEX-cs human TPX2 1–43 | This study | |
| pETSUMO PLK1 3–330 K82R | This study | |
| pETSUMO PLK1 3–330 K82R S99R R106A | This study | |
| pETSUMO Bora 1–120 | This study | |
| pETSUMO Bora 18–120 | This study | |
| pETSUMO Bora 18–120 S59A | This study | |
| pETSUMO Bora 18–120 S112A | This study | |
| pETSUMO Bora 18–120 F56A W58A | This study | |
| pETSUMO Bora 18–120 F56A W58A S112A | This study | |
| pET28a+ Bora 18–120 S27A S41A T52A F56A W58A | This study | |

| Reagent/Resource | Reference or Source | Identifier or Catalog Number |
| --- | --- | --- |
| pET28a+ Bora 18–120 S27A S41A T52A | Genscript | |
| pET28a+ DARPin | Genscript | |
| pETSUMO CEP192 442–533 | Holder et al, 2024 | |
| **Antibodies** | | |
| Anti-Bora | Santa Cruz | Cat #sc-393741 |
| Anti-PLK1 pT210 | Cell Signalling Technologies | Cat #5472 |
| Goat anti-rabbit StarBright 700 secondary | Biorad | Cat #12004162 |
| Goat anti-mouse StarBright 700 secondary | Biorad | Cat #12004159 |
| **Oligonucleotides and other sequence-based reagents** | | |
| Human PLK1 3–330 K82R mutation Forward primer GTTCGCGGGCag-gATTGTGCCTA | IDT | |
| Human PLK1 3–330 K82R mutation Reverse primer ACCTCCTTGGTGTCCGCG | IDT | |
| Human PLK1 3–330 R106A mutation Forward primer ATCCATTCACgc-cAGCCTCGCCC | IDT | |
| Human PLK1 3–330 R106A mutation Reverse primer ATTTCCATAGA-CATCTTCTCC | IDT | |
| Human PLK1 3–330 S99R mutation Forward primer GGAGAAGATGcgtATGGAAA-TATCCATTCACCGCAG | IDT | |
| Human PLK1 3–330 S99R mutation Reverse primer CTCTGGTGCGGCTTGAGC | IDT | |
| Human Bora 18-120 S112A mutation Forward primer GATCGTTCCGGCGCCGTG-GACCG | IDT | |
| Human Bora 18-120 S112A mutation Reverse primer ACGTCTTTGGTAAAGAATTC | IDT | |
| Human Bora 18-120 F56A/ W58A mutation Forward primer tgctAGCATTGAT-CAGCTGGCGGTG | IDT | |
| Human Bora 18-120 F56A/ W58A mutation Reverse primer cgagcCTTGCCCGGGGTCGG-CAG | IDT | |
| Human Bora 18-120 S59A mutation Forward primer GTTTCGTTGGgcgATTGACCA GCTGGCGGTGATCAAC | IDT | |
| Human Bora 18-120 S59A mutation Reverse primer TTGCCCGGGGTCGGC | IDT | |

| Reagent/Resource | Reference or Source | Identifier or Catalog Number |
| --- | --- | --- |
| **Chemicals, Enzymes and other reagents** | | |
| Sodium chloride | Fisher | Cat #BP358-212 |
| ERK2 (2-360) | Dundee reagents | Cat #DU650 |
| ERK1 (2-379) | Dundee reagents | Cat #DU1509 |
| Tris | Melford | Cat #T600-40 |
| Pro-Q Diamond PhosphoProtein gel stain | Thermofisher Scientific | Cat #P33301 |
| TCEP | Fluorochem | Cat #M02624 |
| Tween 20 | Millipore | Cat #655204 |
| Magnesium chloride hexahydrate | Sigma | Cat #63068 |
| Glycerol | Fisher | Cat #G/0650/17 |
| cOmplete™, Mini, EDTA-free Protease Inhibitor Cocktail | Roche | Cat #11836170001 |
| Hepes | Sigma | Cat #H4034 |
| DTT | Melford | Cat #D11000 |
| TBS | Thermo Scientific | Cat #J62938K7 |
| Kanamycin | Sigma | Cat #K4000 |
| Spectinomycin | Thermo Scientific | Cat #J61810.14 |
| Chloramphenicol | Sigma | Cat #C0378 |
| Imidazole | Thermo Scientific | Cat #122025000 |
| IPTG | Protein Arc | Cat #GEN-S-02122 |
| FAM-Bora 52–73 | This study | |
| FAM-Bora F56A W58A 52–73 | This study | |
| FAM-Bora pS59 52–73 | This study | |
| Deuterium oxide | Goss Scientific | Cat #DLM-4 |
| Ammonium chloride ($^{15}$N, 99%) | Goss Scientific | Cat #NLM-467 |
| D-Glucose ($^{13}$C6, 99%) | Goss Scientific | Cat #CLM-1396 |
| Sodium hydrogen phosphate | Acros | Cat #271750025 |
| Potassium dihydrogen phosphate | Sigma | Cat #P3786 |
| Sypro ruby membrane stain | Invitrogen | Cat # S11791 |
| Methanol | Fisher Chemical | Cat #10396090 |
| Glacial Acetic acid | VWR Chemicals | Cat #20104.334 |
| Skim milk powder | Merck | Cat #1153630500 |
| TBS (10x) | Thermo Scientific | Cat #15450277 |
| PBS (10x) | Melford | Cat #P32060 |
| BME vitamin solution (x100) | Sigma-Aldrich/Merck | Cat #B6891 |
| Magnesium sulfate | Sigma | Cat #M7506 |
| Calcium chloride | Sigma | Cat #21097 |
| Iron sulfate heptahydrate | Sigma | Cat #215422 |
| Nickel chloride | Honeywell | Cat #223387 |
| EDTA | Fisher | Cat #BP120 |
| Sodium hydroxide | SLS | Cat #HE3370 |
| All blue standard | Biorad | Cat #1610373 |

| Reagent/Resource | Reference or Source | Identifier or Catalog Number |
|---|---|---|
| Nitrocellulose membrane | Cytiva | Cat #10600000 |
| Ultra Pure ATP | Promega | Cat #V703A |
| Rink amide ProTide resin | CEM corp | Cat #R002-C |
| DMF | Fisher Scientific Ltd | Cat #10284140 |
| Piperidine | Thermofisher Scientific | Cat #A12442.0F |
| DIC | Merck | Cat #38370-500ML |
| OXYMA | Fluorochem Ltd | Cat #F043278-500G |
| 5(6)-Carboxyfluorescein | Fisher Scientific Ltd | Cat #10516081 |
| Triisopropylsilane | Fluorochem Ltd | Cat #S17975-100G |
| DODT | Merck | Cat #465178-100 ML |
| TFA | Fluorochem Ltd | Cat #F008708-1L |
| Fmoc-Ala-OH | Fluorochem Ltd | Cat #M03347-100G |
| Fmoc-Arg(Pbf)-OH | Fluorochem Ltd | Cat #M03398-100G |
| Fmoc-Asn(Trt)-OH | Fluorochem Ltd | Cat #M03352-100G |
| Fmoc-Asp(OtBu)-OH | Fluorochem Ltd | Cat #M03404-100G |
| Fmoc-Gln(Trt)-OH | Fluorochem Ltd | Cat #M03356-100G |
| Fmoc-Glu(OtBu)-OH | Fluorochem Ltd | Cat #M03409-100G |
| Fmoc-Gly-OH | Fluorochem Ltd | Cat #M03361-100G |
| Fmoc-His(Trt)-OH | Fluorochem Ltd | Cat #M03415-100G |
| Fmoc-Ile-OH | Fluorochem Ltd | Cat #M03362-100G |
| Fmoc-Leu-OH | Fluorochem Ltd | Cat #M03365-100G |
| Fmoc-Lys(Boc)-OH | Fluorochem Ltd | Cat #M03419-100G |
| Fmoc-Met-OH | Fluorochem Ltd | Cat #M03368-100G |
| Fmoc-Phe-OH | Fluorochem Ltd | Cat #M03370-100G |
| Fmoc-Pro-OH | Fluorochem Ltd | Cat #M03372-100G |
| Fmoc-Ser(tBu)-OH | Fluorochem Ltd | Cat #M03382-100G |
| Fmoc-Thr(tBu)-OH | Fluorochem Ltd | Cat #M03389-100G |
| Fmoc-Trp(Boc)-OH | Fluorochem Ltd | Cat #M03376-100G |
| Fmoc-Tyr(tBu)-OH | Fluorochem Ltd | Cat #M03428-100G |
| Fmoc-6-Aminohexanoic acid | Fluorochem Ltd | Cat #F045380-25G |

| Reagent/Resource | Reference or Source | Identifier or Catalog Number |
|---|---|---|
| Fmoc-Ser(HPO3Bzl)-OH | Fluorochem Ltd | Cat #M03387-5G |
| 1-Methyl-2-pyrrolidone | Merck | Cat #M79603-1L |
| Acetic anhydride | Merck | Cat #8222781000 |
| Acetonitrile | Merck | Cat #34851-2.5L |
| Whatman 540, 185 mm circles | Cytiva | Cat # 1540-185 |
| Pro-Q™ Diamond Phosphoprotein Gel Stain | ThermoFisher Scientific | Cat # P33301 |
| **Software** | | |
| Origin | OriginLab https://www.originlab.com/ | |
| AlphaFold3 | https://www.alphafoldserver.com | |
| AlphaFold2 | https://github.com/sokrypton/ColabFold | |
| AlphaBridge | https://alpha-bridge.eu/ | |
| PDBe PISA | https://www.ebi.ac.uk/pdbe/pisa/ | |
| KinCoRe | http://dunbrack.fccc.edu/kincore/home | |
| PSI-Blast | https://blast.ncbi.nlm.nih.gov/Blast.cgi | |
| MAFFT | https://mafft.cbrc.jp/alignment/software/ | |
| Uniprot | https://www.uniprot.org/ | |
| NMRpipe/NMRDraw | https://doi.org/10.1007/BF00197809 | |
| CCPNMR Analysis v2.5 | https://doi.org/10.1002/prot.20449 | |
| GraphPad Prism | www.graphpad.com | |
| PLGS (v3.0.2) | | |
| DynamX (v3.0.0) | | |
| Deuteros 2.0 | https://github.com/andymlau/Deuteros_2.0 (Lau et al, 2021) | |
| iBright image analysis software | https://www.thermofisher.com/uk/en/home/life-science/protein-biology/protein-assays-analysis/western-blotting/detect-proteins-western-blot/western-blot-imaging-analysis/ibright-systems/software.html#ibright-analysis-software-secure | |
| **Other** | | |
| Hidex Sense plate reader | Hidex | |
| Akta pure protein purification system | Cytiva | |

| Reagent/Resource | Reference or Source | Identifier or Catalog Number |
|---|---|---|
| 5 ml HisTrap HP | Cytiva | Cat #17524802 |
| Amicon Ultra Centrifuge Filters 10 kDa MWCO | Millipore | Cat #UFC901024 |
| Q5 Mutagenesis kit | NEB | Cat #E0554S |
| Quikchange SDM kit | Aligent | Cat #200519 |
| HiLoad Superdex 200 pg SEC column | Cytiva | Cat #28989335 |
| Superose 12 10/300 GL | Cytiva | Cat #GE17-5173-01 |
| Pur-A-lyzer mini dialysis kit | Sigma | Cat #PURD35050 |
| SnakeSkin™Dialysis Tubing | Thermo Fisher Scientific | Cat # 68100 |
| Low volume non-binding 384 well black plate | Greiner | Cat #784900 |
| 5 mm NMR tube | Norell | |
| 5 mm Shigemi NMR tube | Shigemi | |
| 750 MHz Oxford Instruments Spectrometer, 5 mm Bruker TCI cryoprobe, Bruker Avance III HD console | Bruker | |
| Liberty Blue pep. synthesizer | CEM corp. | |
| MultiPep 2 | CEM corp. | |
| Agilent 1260 infinity HPLC | Agilent | |
| Agilent 1290 Infinity II HPLC | Agilent | |
| Kinetex EVO 5 µm C18 100 Å 21.2 × 250 mm RP column | Phenomenex | |
| maXis II™ Impact QToF | Bruker | |

## Reagents

Human His-GST tagged ERK2 (2-360) and ERK1 (2-379) were purchased from MRC PPU Reagents and Services (DU650, DU1509). Human Bora 18–120, 18–120 S27A S41A T52A and PLK1 binding DARPin were purchased as codon optimised sequences from GenScript. Oligonucleotides used for site-directed mutagenesis and subcloning were ordered from IDT. Site-directed mutagenesis was performed using the Q5 site-directed mutagenesis kit (NEB). The constructs used in this publication are listed in the reagents and tools table. All DNA constructs were verified by sequencing. Pro-Q Diamond PhosphoProtein gel stain was purchased from ThermoFisher Scientific.

The antibodies used in this study are: anti-Bora (Santa Cruz Ca. sc-393741), antiPhospho-Plk1 pT210 (Cell Signalling Technologies Cat. No. 5472), goat anti-rabbit StarBright 700 secondary (Biorad Cat. No. 12004162), goat anti-mouse StarBright 700 secondary antibody (Biorad Cat. No. 12004159).

## Protein production

Bora 18–120, 1–120 and mutants were cloned into the petSUMO vector with an N-terminal TEV protease cleavable His-SUMO tag.

This was transformed into B834 RIL cells. Four litres of cells were inoculated with 10 ml of overnight culture. Protein expression was induced overnight with 0.5 mM IPTG at 20 °C. The pellet was resuspended in 10 ml per litre of ice-cold lysis buffer (50 mM TRIS pH 7.5, 500 mM NaCl, 10% glycerol, 0.5 mM TCEP, 20 mM imidazole, EDTA-free protease inhibitors). The cells were sonicated for 10 s on, 20 s off for 4 min 10 s total at 60%. The soluble fraction was collected at 17000 rpm for 45 min. This was then filtered through 0.45 µm filters before loading onto a HisTrap FF. The bound protein was eluted in a gradient of 500 mM imidazole. This was dialysed overnight into 250 mM NaCl, 50 mM TRIS pH 7.5, 1 mM TCEP, 10% glycerol in the presence of TEV protease. The following morning the protein was incubated with 3 ml of Ni-NTA equilibrated in dialysis buffer on the roller for 45 min to bind any uncut protein or His-SUMO. The flowthrough containing Bora was concentrated and loaded onto an SD200 16/600 size exclusion column equilibrated in 300 mM NaCl, 50 mM TRIS pH 7.5, 10% glycerol, 1 mM TCEP. The clean fractions were concentrated in a 10 kDa cut off concentrator and the protein was flash-frozen and stored at −80 °C.

Human PLK1 3–330 and mutants were cloned into the petSUMO vector with an N-terminal TEV protease cleavable His-SUMO tag. This was transformed into B834 RIL cells. Four litres of cells were inoculated with 10 ml of overnight culture. Protein expression was induced overnight with 0.5 mM IPTG at 20 °C. The pellet was resuspended in 10 ml per litre of ice-cold lysis buffer (50 mM TRIS pH 7.5, 500 mM NaCl, 10% glycerol, 0.5 mM TCEP, 20 mM imidazole, 5 mM $MgCl_2$, EDTA-free protease inhibitors). The cells were sonicated for 10 s on, 20 s off for 4 min 10 s total at 60%. The soluble fraction was collected at 17,000 rpm for 45 min. This was then filtered through 0.45 µm filters before loading onto a HisTrap HP. The bound protein was eluted in a gradient of 500 mM imidazole. This was dialysed overnight into 250 mM NaCl, 50 mM TRIS pH 7.5, 1 mM TCEP, 10% glycerol, 5 mM $MgCl_2$ in the presence of TEV protease to remove the His-SUMO tag. The following morning the protein was loaded onto a HisTrap HP equilibrated in dialysis buffer, and the flow through collected. Bound protein was eluted in a gradient of imidazole after washing with 4 CVs of dialysis buffer. The PLK1 partly eluted in the flow through and partly in the start of the imidazole gradient. This was concentrated in a 30 kDa cut off concentrator and loaded onto the SD200 16/600 size exclusion column equilibrated in 300 mM NaCl, 50 mM TRIS pH 7.5, 10% glycerol, 1 mM TCEP, 5 mM $MgCl_2$. The clean fractions were concentrated in a 30 kDa cut off concentrator and the protein was flash-frozen and stored at −80 °C.

Human Aurora-A kinase domain 122–403 and mutants in an N-terminal His-tagged vector (pET30TEV) were transformed into RIL cells alongside the pCDF vector encoding lambda phosphatase. The bacteria were grown in LB at 37 °C until the O.D. at 600 nm reached 0.6–0.8. Expression was then induced with 0.5 mM IPTG overnight at 20 °C. The pelleted cells were resuspended in 10 ml of ice-cold lysis buffer per litre of culture (50 mM TRIS pH 7.5, 250 mM NaCl, 20 mM imidazole, 10% glycerol, 5 mM $MgCl_2$, one EDTA-free protease inhibitor tablet per 50 ml of buffer). The resuspended cells were sonicated at 60% amplitude for 10 s on, 20 s off, 5 min total to lyse them. The soluble fraction was collected at 17,000 rpm for 5 min. After filtering through a 0.45 µm filter the soluble was loaded onto a HisTrap HP column, washed and eluted in lysis buffer using a gradient of maximum 500 mM imidazole.

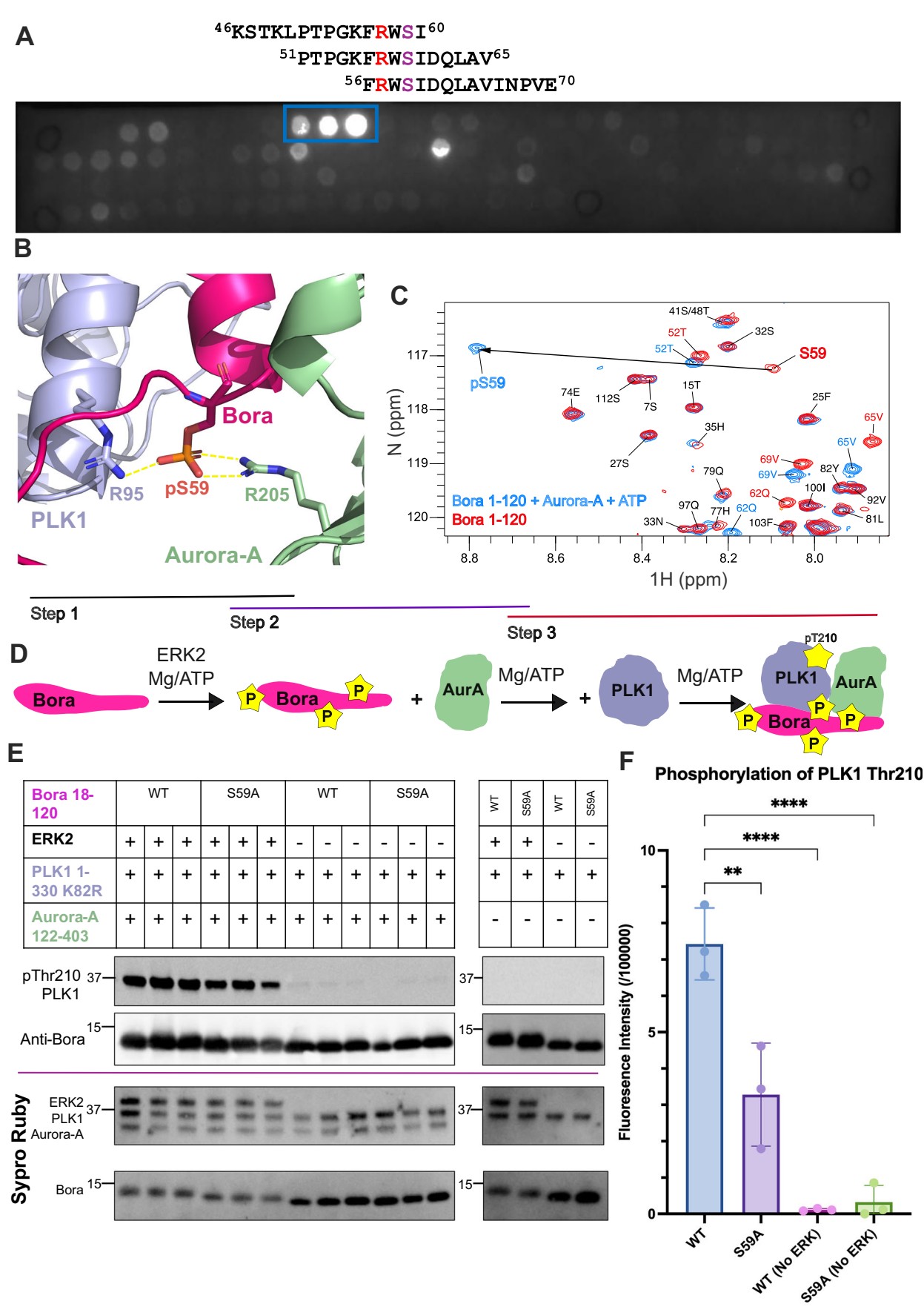

Figure 7. Aurora-A phosphorylated Bora at Ser59.

(A) Peptide array of Bora divided into peptides of 15 amino acids. The array was incubated with ATP and active Aurora-A kinase domain before probing with Pro-Q Diamond Phosphoprotein stain and imaging. Clear phosphorylation is seen on the peptides that include Ser59 of Bora. (B) Modelling of pSer59 in the ternary complex of Bora (magenta), Aurora-A (green) and PLK1 (blue). The phosphorylated residue is in the interface between PLK1 and Aurora-A. Interactions are predicted between the phosphate on Ser59 and Arg95 in PLK1 and Arg205 in Aurora-A. (C) $^1$H–$^{15}$N HSQC spectrum of $^{15}$N$^{13}$C-labelled human Bora 1–120 in the absence (red) and presence (blue) of active Aurora-A kinase domain and ATP. A clear shift in the position of the Ser59 peak to the left indicates that this is it being modified by Aurora-A in vitro. (D) Altered assay schematic where Aurora-A wild-type kinase domain is preincubated with phosphorylated Bora 18–120 wild-type and S59A before the PLK1 K82R kinase domain is included as a substrate. (E) Western blot analysis of the levels of phosphorylation of PLK1 kinase domain at Thr210 when Bora 18–120 wild-type and S59A are preincubated with ERK2, followed by Aurora-A before addition to PLK1 K82R kinase domain. Blots were probed with antibodies to Bora and pThr210 in PLK1. The samples were repeated in triplicate to allow quantification. Inputs are shown with Sypro Ruby staining of nitrocellulose membranes. (F) Quantification of the levels of phosphorylation of PLK1 at Thr210 in the presence of either wild-type Bora 18–120 or S59A Bora, with and without incubation with ERK2. Data represents three independent technical replicates; the measure of the centre of the error bars is the mean, with the error bars denoting ± standard deviation (SD). ** indicates an adjusted $p$-value of 0.0012, **** indicates $p < 0.0001$ with analysis in comparison to WT Bora 18–120 by one-way ANOVA. $n = 3$. Source data are available online for this figure.

The His-tag was then cleaved overnight using TEV protease in dialysis at 4 °C into 50 mM TRIS pH 7.5, 250 mM NaCl, 10% glycerol, 5 mM MgCl$_2$, 1 mM TCEP. After dialysis the cleaved protein was rebound to the HisTrap equilibrated in dialysis buffer and a gradient of 500 mM imidazole was used to elute the tag-free protein. The Aurora-A-containing fractions were concentrated in a 10 kDa cut-off concentrator and loaded onto a SD200 16/600 size exclusion column equilibrated into 50 mM TRIS pH 7.5, 200 mM NaCl, 10% glycerol, 5 mM MgCl$_2$, 1 mM TCEP. In the final step Aurora-A was concentrated again and flash-frozen before storage at −80 °C.

PLK1-binding DARPin in an N-terminal His-tagged vector were transformed into B834 RIL cells. The bacteria were grown in LB at 37 °C until the O.D. at 600 nm reached 0.6–0.8. Expression was then induced with 0.5 mM IPTG overnight at 20 °C. The pelleted cells were resuspended in 10 ml of ice-cold lysis buffer per litre of culture (50 mM TRIS pH 7.5, 500 mM NaCl, 20 mM imidazole, 10% glycerol, 5 mM MgCl$_2$, one EDTA-free protease inhibitor tablet per 50 ml of buffer). The resuspended cells were lysed by sonication at 60% amplitude for 10 s on, 20 s off, 5 min total. The soluble fraction was collected at 17,000 rpm for 5 min. After filtering through a 0.45 μm filter the soluble fraction was loaded on a HisTrap HP column. The DARPin was eluted in lysis buffer with a gradient of imidazole up to 500 mM maximum, concentrated to under 5 ml in a 5 kDa cut-off concentrator and loaded onto a SD200 16/600 column equilibrated into 50 mM TRIS pH 7.5, 250 mM NaCl, 10% glycerol, 5 mM MgCl$_2$, 1 mM TCEP. In the final step the DARPin was concentrated again and flash-frozen before storage at −80 °C.

CEP192 442–533 for use in competition NMR was expressed and purified as detailed in previous work (Holder et al, 2024). TPX2 1–43 for use in competition NMR was expressed and purified as detailed in previous work (McIntyre et al, 2017).

## NMR

For NMR studies Bora 1–120, Bora 18–120 S27A S41A T52A, and Bora 18–120 F56A W58A were expressed in BL21 (DE3) *E. coli* in 250 mL of $^{15}$N/$^{13}$C minimal media as His-SUMO-Bora fusions (Holder et al, 2024; Rejnowicz et al, 2024). Minimal media contained 2 g/L $^{15}$NH$_4$Cl and 4 g/L $^{13}$C D-glucose in 50 mM Na$_2$HPO$_4$, 25 mM KH$_2$PO$_4$, 20 mM NaCl, supplemented with 2 mM MgSO$_4$, 0.2 mM CaCl$_2$, 0.01 mM FeSO$_4$, a micronutrient cocktail and a vitamin solution (BME vitamins 100× solution,

Sigma-Aldrich). Proteins were purified from cell lysate using His-tag affinity chromatography in TRIS buffers (50 mM TRIS, 250/350 mM NaCl, 20 to 500 mM imidazole, 10% glycerol, pH 7.5). The His-SUMO tag was cleaved with TEV and separated by rebinding to the HisTrap column. BORA proteins were polished using size exclusion chromatography into 'NMR buffer': 20 mM (K/H)$_3$PO$_4$ 150 mM NaCl, 2.5% glycerol, pH 6.8. TEV cleavage leaves a short 'hangover' sequence GS or GSM at the N-terminus.

The following NMR spectra were recorded at 10 °C with a 0.3 mM sample of $^{15}$N/$^{13}$C-labelled Bora 1–120: $^1$H–$^{15}$N HSQC, $^1$H–$^{13}$C HSQC, HNCO, HNCA, HNCoCA (recorded uniformly), and HNcaCO, HNCACB and HNcocaCB (recorded with NUS). These spectra allowed the complete ab initio assignment of the $^1$H–$^{15}$N HSQC spectrum and all accessible HN, H, CO, Cα, and Cβ resonances were recorded. An HBHAcoNH spectrum in combination with the $^1$H–$^{13}$C HSQC provided assignments for all but a handful of the Hα and Hβ protons. Secondary shifts ($\Delta\delta = \delta - \delta_{ref}$) were calculated for CO, Cα, and Cβ nuclei with reference coil values generated using a web server hosted by the University of Copenhagen (Hendus-Altenburger et al, 2019).

Residue-specific helical propensities within Bora 1–120 were calculated using a TALOS-N analysis of all backbone chemical shifts (Shen and Bax, 2013).

Additional $^1$H–$^{15}$N-HSQC spectra were recorded at 15, 20, 25 and 30 °C. Transverse relaxation rates ($R_2$) and heteronuclear NOEs (hetNOE) for backbone $^{15}$N nuclei were measured at 10 °C. Recycle delays were 2.5 s for $R_2$ and 5.0 s for hetNOE experiments. For $R_2$, twelve relaxation periods were used ranging from 17 to 204 ms; two were duplicated to help with error estimations. Measurements were limited to those residues with independent, resolvable peaks in the $^1$H–$^{15}$N HSQC spectrum. Peak intensities were measured and relaxation rates/hetNOE values analysed using PINT (Ahlner et al, 2013).

$^1$H–$^{15}$N HSQC assignments for a 0.3 mM sample of Bora 18–120 S27A S41A T52A and a 0.2 mM sample of Bora 18–120 F56A W58A were achieved through direct comparison with Bora 1–120 spectra and by using a HNCA/HNCoCA pair of spectra to confirm assignments at the N-terminus and close to mutation sites.

## Bora phosphorylation in NMR

Wild-type Aurora-A 122–403 (final conc. ~1 μM) was added to 30–80 μM samples of $^{15}$N/$^{13}$C-labelled Bora 1–120 or Bora 18–120 S27A S41A T52A in NMR buffer supplemented with 1–2 mM ATP

and 5 mM MgCl$_2$ at 20–25 °C. The phosphorylation status after addition of kinase was tracked by recording sequential $^1$H–$^{15}$N HSQC spectra. The identity of pSer59 was confirmed using an HNCA experiment at 10 °C, linking through ($i,i-1$) Cα resonances for shifted peaks. Clear $^1$H–$^{15}$N chemical shift perturbations (>0.02 ppm) were shown for residues between Thr52 and Ile71 (inclusive).

Similarly, for ERK phosphorylation, ERK1 or ERK2 (final conc. ~1 μM) was added to 50 μM $^{15}$N/$^{13}$C-labelled samples of Bora 1–120 or Bora 18–120 S27A S41A T52A or Bora 18–120 F56A W58A in NMR buffer supplemented with 1 mM ATP and 3 mM MgCl$_2$ at 20–25 °C. The identity of the phosphorylated residues (pSer112 only for Bora 18–120 S27A S41A T52A) was confirmed using an HNCA experiment or HNCA/HNcoCA pair at 10 °C. When required for subsequent binding studies, further phosphorylation was quenched by sequestering Mg$^{2+}$ ions through addition of equimolar EDTA (0.1 M stock). Phosphorylation of Bora 18–120 F56A W58A was stopped once the peak intensities for Ser112 and pSer112 (and Thr115 in both states) were roughly equal.

## NMR titrations/competition assays

For binding studies, small aliquots of concentrated Aurora-A 122–403 C290A/C393A or dephosphorylated WT Aurora-A 122–403 were added to 30–80 μM samples of isotopically labelled Bora 1–120, Bora 18–120 S27A S41A T52A (unphosphorylated, pSer112 or pSer59) or partially Ser112-phosphorylated Bora 18–120 F56A W58A in NMR buffer up to a maximum 1:1 molar ratio.

For competition assays, Aurora-A was first added up to a 1:2.5 molar ratio of [AurA]:[Bora] and then TPX2 1–43 or CEP192 442–533 was added in small, concentrated aliquots up to a final molar ratio of 1:2.5:1. Peak intensities were ratioed to the original Bora-only spectrum; the change in Bora concentration was small.

Estimates of the error ($\Delta R$) for peak intensity ratios between two spectra ($R$) were determined from the baseline noise. The NMPpipe command 'showApod' was used to determine the noise in spectrum 1 ($\Delta z_1$) and spectrum 2 ($\Delta z_2$). Errors were propagated using ($\Delta R/R$)$^2$ = ($\Delta z_1/z_1$)$^2$ + ($\Delta z_2/z_2$)$^2$, where $z_1$ and $z_2$ are peak intensities in each spectrum.

## PLK1 pThr210 phosphorylation assay

To create phosphorylated Bora, Bora WT and mutants were preincubated at 5 μM with 640 nM ERK2 in 50 μl of phosphorylation buffer (25 mM HEPES pH 7.5, 150 mM NaCl, 1 mM DTT, 20 mM MgCl$_2$, 200 μM ATP) for 2 h at 30 °C. The phosphorylated Bora at a final concentration of 1 μM was then mixed with 200 nM of PLK1 K82R 3–330 and Aurora-A 122–403 in 30 μl total phosphorylation buffer. This was incubated at 30 °C for 30 min before 30 μl of 2 X SDS loading dye was added, the samples were boiled and were analysed via western blot with transfer onto a nitrocellulose membrane. The blots were blocked in 5% milk in TRIS buffered saline with 0.01% Tween 20. The membranes were split and probed with anti-Bora and anti-PLK1 pT210 antibodies. Fluorescent secondary antibodies from Biorad were used and the western blot imaged using the iBright imaging system (Invitrogen). The iBright analysis software was used to quantify the levels of fluorescence using the local background corrected volume. When analysing the effects of Ser59 phosphorylation in Bora, there was an

extra 30 min incubation period of the phosphorylated Bora with Aurora-A before the PLK1 was added.

The levels of all proteins included in the phosphorylation assay were assessed using the SYPRO Ruby staining of the nitrocellulose membrane (SYPRO Ruby protein blot stain, Invitrogen), with visualisation on the iBright imaging system (Invitrogen).

## Modelling of ternary complexes

The Google Colab version of AlphaFold2 multimer in local mode was used to model a three-way complex between human Bora (Q6PGQ7), human PLK1 (P53350) and human Aurora-A (O14965) (Mirdita et al, 2022) (preprint: Evans et al, 2021). 10 seeds were used in the modelling with 6 recycles. The top ipTM score from the 50 models is included in Appendix Table S1. Modelling the complexes between Aurora-A kinase, PLK1 and a phosphorylated version of Bora was achieved using the alphafold-server.com and AlphaFold3 (Abramson et al, 2024). The full-length sequences of Aurora-A, Bora and PLK1 were taken from Uniprot. Models used in this study were deposited to ModelArchive (Tauriello et al, 2025).

Pairwise ipTM scores were extracted from the json output from alphafoldserver.com. 10 models were generated for each complex, with all 5 ipTM scores extracted from each model and plotted on a violin plot using Origin (OriginPro version 2024). AlphaBridge was used to identify the predicted interfaces between protein complexes modelled on AlphaFold3 without ADP or phosphorylation present (preprint: Álvarez-Salmoral et al, 2024).

## Phylogenetic analysis

Orthologous sequences were identified using iterative PSI-BLAST searches on the non-redundant database using the UniProt sequence Q6PGQ7 Bora 1–559, P53350 PLK1 1–603 or O14965 Aurora-A 1–403 as a template. Diverse orthologues were identified by constraining the searches to use just a particular taxonomic group. The multiple sequence alignments of PLK1 and Bora were generated using MAFFT (Katoh et al, 2017). Sequence conservation logos were created using the MAFFT alignment and WebLogo (Crooks et al, 2004). The *Strongylocentrotus purpuratus* orthologues used in the AlphaFold3 modelling were XP_030849245.1 (PLK1, 1–330), XP_030831330.1 (Aurora-A 80–346) and XP_030848668.1 (Bora 18–112). The *Drosophila melanogaster* orthologues used in the AlphaFold3 modelling were Bora (Q9VVR2 72–165), Aurora-A kinase (Q9VGF9 151–411) and PLK1 (P52304 21–280).

## Analytical SEC

A superose 12 10/300 column (Cytiva) was equilibrated in 50 mM TRIS pH 7.5, 250 mM NaCl, 10% glycerol, 5 mM MgCl$_2$, 0.5 mM TCEP. PLK1 3–330 K82R or K82R R106A S99R and DARPin were mixed at 1:1 molar ratio in a total volume of 500 μl. The mixed sample and the individual proteins were subject to SEC in sequential runs and the resulting traces overlaid.

## Fluorescence anisotropy-based direct binding assays

PLK1 (3–330 K82R or K82R R106A S99R) was buffer exchanged into the assay buffer (25 mM HEPES pH 7.5, 50 mM NaCl, 2 mM

DTT, 5 mM MgCl$_2$, 0.01% Tween 20), then diluted across a 384 low-volume black plate (Greiner) from high to low concentration, before the addition of 50 nM tracer (FAM-Bora 52–73 wild-type, FAM-Bora pSer59 52–73 or FAM-Bora 52–73 F56A W58A) to three replicate rows, whereas only buffer was added to the control row. Plates were left incubating for 1 h at room temperature before the fluorescence anisotropy was measured using a HIDEX plate reader, with excitation at 485 nm and emission at 535 nm.

Fluorescence anisotropy data were processed using Microsoft Excel to calculate intensity and anisotropy using the equation listed in Yeo et al (2013). Data was fitted on Origin (OriginPro version 2025).

## Peptide synthesis

Peptide synthesis was performed using a Liberty Blue peptide synthesiser (CEM Corporation) with microwave heating at 0.1 mmol scale on Rink Amide ProTide resin (loading 0.19 mmol/g). Standard preprogrammed coupling and deprotection cycles were applied. The deprotection was achieved using 20% piperidine in DMF with microwave heating at 90 °C for 100 s, followed by three DMF washing steps. The couplings were performed using 5 eq. of Fmoc-protected amino acid, 5 eq. of N,N'-diisopropylcarbodiimide (DIC) and 5 eq. of 2-cyano-2-(hydroxyimino)acetate (Oxyma) in DMF with microwave heating at 90 °C for 3 min followed by two DMF washing steps.

The resin with synthesized peptide was then transferred to a SPS tube and labelled using 3 eq. of 5(6)-carboxyfluorescein, 3 eq. of DIC and 3 eq. of Oxyma in DMF for 16 h, followed by washing with 10 ml of 20% piperidine in DMF two times for 5 min and three times with 10 ml DMF. After further washing three times with 10 ml dichloromethane and two times with 10 ml of diethylether the resin was dried under vacuum for 30 min. To deprotect the side chains and cleave the peptide from the resin, the resin was incubated on a rotator for 3 h with 10 ml of cleavage mix (92.5% trifluoroacetic acid (TFA), 2.5% water, 2.5% triisopropylsilane (TIPS), 2.5% 3,6-dioxa-1,8-octanedithiol (DODT) and filtered. The filtrate was concentrated to ca 1 ml under a stream of nitrogen and the peptide was precipitated by addition of 10 ml of ice-cold diethylether and isolated by centrifugation (6000 rpm for 5 min). The precipitate was resuspended in 10 ml of ice-cold diethylether and isolated by repeating the centrifugation step. After decanting the diethylether, the precipitate was allowed to dry for 30 min, dissolved in 5 ml of 1% (v/v) acetic acid and freeze dried.

## Peptide purification

Peptide was dissolved in 10 ml of 1:1 mixture of acetonitrile and water and purified using an Agilent 1260 infinity system equipped with UV detector and fraction collector on a Kinetex EVO 5 μm C18 100 Å 21.2 × 250 mm reverse phase column. 2.5 ml of the peptide solution was injected and a 25 min gradient of 20–40% acetonitrile in water with 0.1% formic acid additive was run at 10 ml/min. The fractions containing peptide were pooled and freeze dried.

The identity of the peptides was confirmed by high-resolution mass spectrometry on an Impact II QTOF spectrometer (Bruker) using electrospray ionisation. The purity was determined by analytical HPLC on an Agilent 1290 Infinity II system using an Ascentis peptide column and a 5–95% gradient of acetonitrile in water with 0.1% trifluoroacetic acid additive at 0.5 ml/min for 10 min.

## Mass spectrometry analysis

Bora 18–120 wild-type and mutants were diluted to 8 μM in 25 mM HEPES pH 7.5, 150 mM NaCl, 20 mM MgCl$_2$, 1 mM DTT and 200 μM ATP with and without inclusion of 1.5 μM ERK2. After incubation at 30 °C for 2 h, EDTA was added to a final concentration of 20 mM to stop any further phosphorylation. The samples were flash-frozen and stored ahead of analysis.

Accurate mass spectra were acquired on an Impact II QTOF spectrometer (Bruker) using electrospray ionisation. Samples were introduced using an HTC PAL autosampler and Bruker Elute Pump. The HPLC column (Waters Acquity Vanguard Protein BEH C4 300 Å 1.7 μm, 2.1 mm × 100 mm with a 1.7 μm 2.1 mm × 5 mm precolumn) was heated to 40 °C. The separation was achieved using a 5–95% gradient of acetonitrile in water with 0.1% formic acid additive. Maximum entropy deconvolution methods were used as part of the processing.

## Peptide array

Peptide arrays representing human Bora 1–559 as 15-mer peptides with a 5 residue shift were synthesized on 10 × 15 cm cellulose membrane made of 6-aminohexanoic acid modified Whatman 540 filter paper, using a MultiPep 2 peptide synthesizer (CEM Corporation). First, Fmoc-8-amino-3,6-dioxaoctanoic acid spacer was coupled to the membrane to provide a distance between the solid support and the peptide. Peptides were then assembled using standard Fmoc-based solid phase synthesis with double deprotection step (2 × 15 min) using 20% piperidine in DMF, double couplings (2 × 30 min) with DIC and Oxyma as coupling reagents and 10% acetic anhydride in DMF as capping reagent (15 min). Final deprotection was achieved by incubating the membrane in the mixture of 92.5% TFA, 2.5% water, 2.5% TIPS and 2.5% DODT for 3 h, followed by three dichloromethane, three DMF, three ethanol washes and air-drying.

The peptides on the membrane were rehydrated in ethanol for 5 min followed by a 5 min wash in TBST (50 mM TRIS pH 7.5, 150 mM NaCl, 0.01% Tween-20). The membrane was blocked with 1% bovine serum albumin (BSA) in phosphorylation buffer (25 mM TRIS pH 7.5, 150 mM NaCl, 5 mM MgCl$_2$, 1 mM DTT, 0.01% Tween-20, 10% glycerol) with agitation for 1 h and washed for 5 min in binding buffer. The membrane was then incubated with 100 nM of autophosphorylated Aurora-A 122–403 and 100 μM ATP in phosphorylation buffer for 1 h for phosphorylation to occur. The protein was drained off and the membrane washed using 10% SDS (2 × 5 min) and deionised water (5 × 5 min). The water was drained off and the membrane incubated in Pro-Q Diamond Phosphoprotein stain (ThermoFisher Scientific) for 1 h. The stain was removed and the membrane washed with deionised water (2 × 5 min). and destained with 50 mM NaOAc pH 4.0 + 5% acetonitrile (2 × 15 min). The level of phosphorylation was then imaged using a ChemiDoc MP imaging system (BioRad).

# Data availability

The datasets produced in this study are available in the following databases: NMR assignment for Bora 1–120: Biological Magnetic Resonance Data Bank (52970). AlphaFold molecular models:

ModelArchive PLK1 21–330, Bora 18–120 with Aurora-A 122–403 with AlphaFold3 ma-1klkx; Full-length PLK1, Bora and Aurora-A with AlphaFold2 ma-uptw5; PLK1 21–330, Bora 18–120 with Aurora-A 122–403 with AlphaFold2 ma-x9q8r; Full-length PLK1, Bora and Aurora-A with AlphaFold3 ma-bxipa; PLK1 21–330, Bora 18–120 with Aurora-A 122–403 with AlphaFold3 including phosphorylation of Bora Ser59 and Ser112 ma-bldfw. Mass spectrometry analysis of Bora phosphorylation: BioStudies S-BSST2315.

The source data of this paper are collected in the following database record: biostudies:S-SCDT-10_1038-S44319-025-00687-z.

# Peer review information

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

## Acknowledgements

This research was supported by the Biotechnology and Biological Sciences Research Council (BBSRC) (BB/V003577/1 and BB/V003577/2). We thank Arnout Kalverda for NMR support and Roland Dunbrack Jr (Fox Chase Cancer

Centre) for help with modelling. Facilities for NMR spectroscopy were funded by the University of Leeds (ABSL award) and Wellcome Trust (108466/Z/15/Z).

## Author contributions

**Jennifer A Miles**: Conceptualization; Formal analysis; Validation; Investigation; Visualization; Methodology; Writing—original draft; Writing—review and editing. **Matthew Batchelor**: Formal analysis; Investigation; Visualization; Methodology; Writing—original draft; Writing—review and editing. **Martin Walko**: Resources; Formal analysis; Investigation; Visualization; Writing—original draft. **Vanda Gunning**: Investigation. **Andrew J Wilson**: Conceptualization; Supervision; Funding acquisition; Writing—review and editing. **Megan H Wright**: Conceptualization; Supervision; Funding acquisition; Writing—review and editing. **Richard Bayliss**: Conceptualization; Supervision; Funding acquisition; Writing—original draft; Writing—review and editing.

Source data underlying figure panels in this paper may have individual authorship assigned. Where available, figure panel/source data authorship is listed in the following database record: biostudies:S-SCDT-10_1038-S44319-025-00687-z.

## Disclosure and competing interests statement

The authors declare no competing interests.

# Expanded View Figures

**Figure EV1.  Modelling the ternary complex between Bora, PLK1 and Aurora-A.**

(A) Model of the complex of full-length human Bora (bright pink) bound to full-length Aurora-A (green) and full-length PLK1 (blue) from AlphaFold3. (B) Model of the complex of full-length human Bora bound to full-length Aurora-A and full-length PLK1 from AlphaFold3. The model is coloured according to the pLDDT scores of each residue in the modelling, with most of the model containing intrinsically disordered regions of low confidence. The blue regions have the highest confidence, and the orange regions have the lowest confidence. (C) Simplified model removing the low confidence disordered regions, with Bora shown in magenta, Aurora-A shown in green and PLK1 shown in blue. The short Bora peptide covering 245–257 is modelled interacting with the polo-box domain of PLK1. (D) AlphaBridge summary of the predicted interfaces between full-length human Aurora-A and Bora from an AlphaFold3 model. The colouring is based on the pLDDT score. (E) AlphaBridge summary of the predicted interfaces between full-length Aurora-A, PLK1 and pS112 Bora from an AlphaFold3 model. The colouring is based on the pLDDT score.

▶

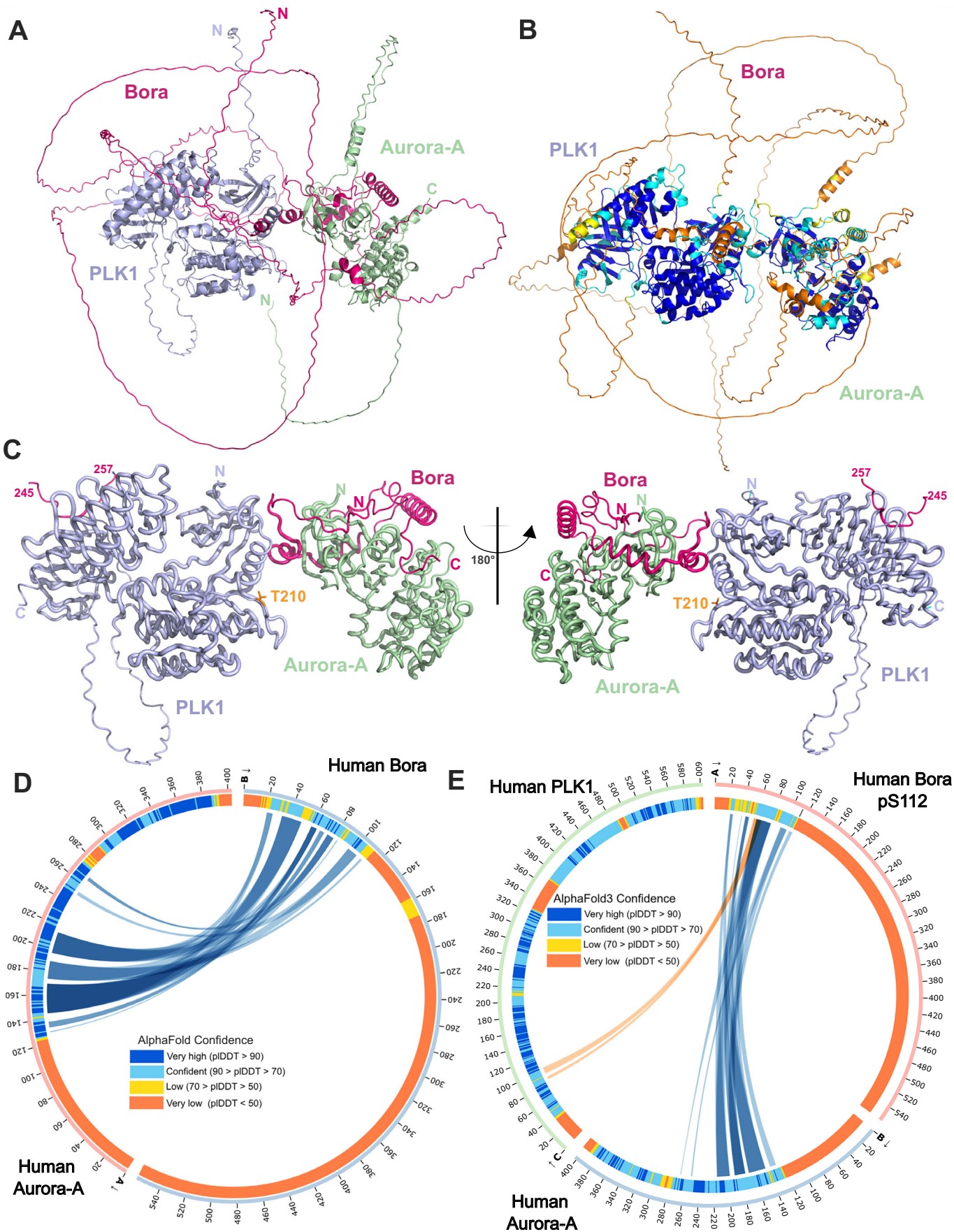

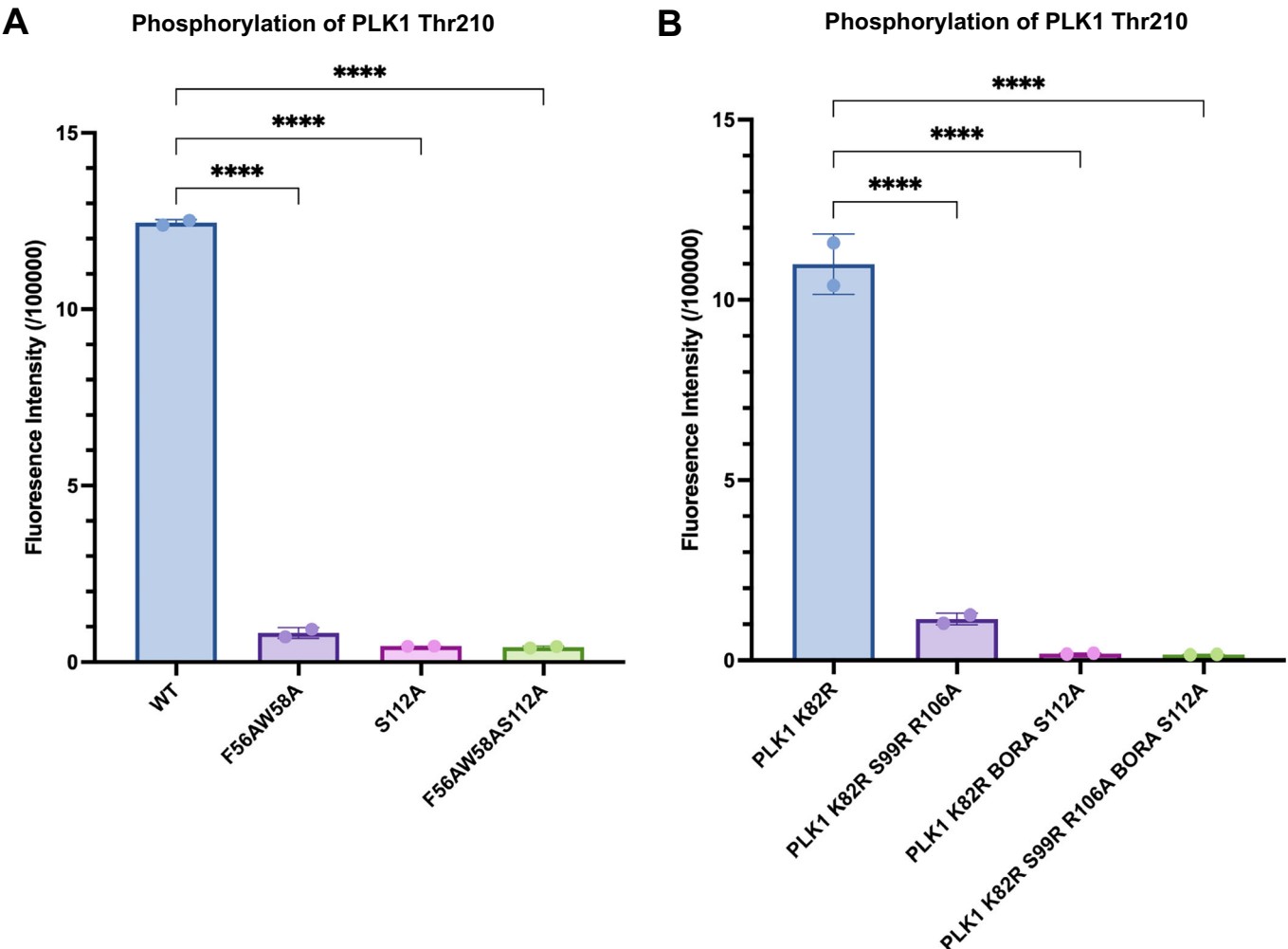

**Figure EV2. Quantification of PLK1 Thr210 phosphorylation in Fig. 2.**

(A) Quantification of the levels of PLK1 phosphorylation on Thr210 in Fig. 2D, with the version of Bora 18–120 shown on the x-axis. iBright analysis software was used to quantify the levels of fluorescence in the bands shown. Data represents two independent technical replicates, with the mean plotted and the standard deviation shown as error bars. (B) Quantification of the levels of PLK1 phosphorylation on Thr210 in Fig. 2E. The different Bora 18–120 and PLK1 3–330 proteins used in each of the assays are listed below. iBright analysis software was used to quantify the levels of fluorescence in the bands shown. Data represents two independent technical replicates, with the mean plotted and the standard deviation shown as error bars. Source data are available online for this figure.

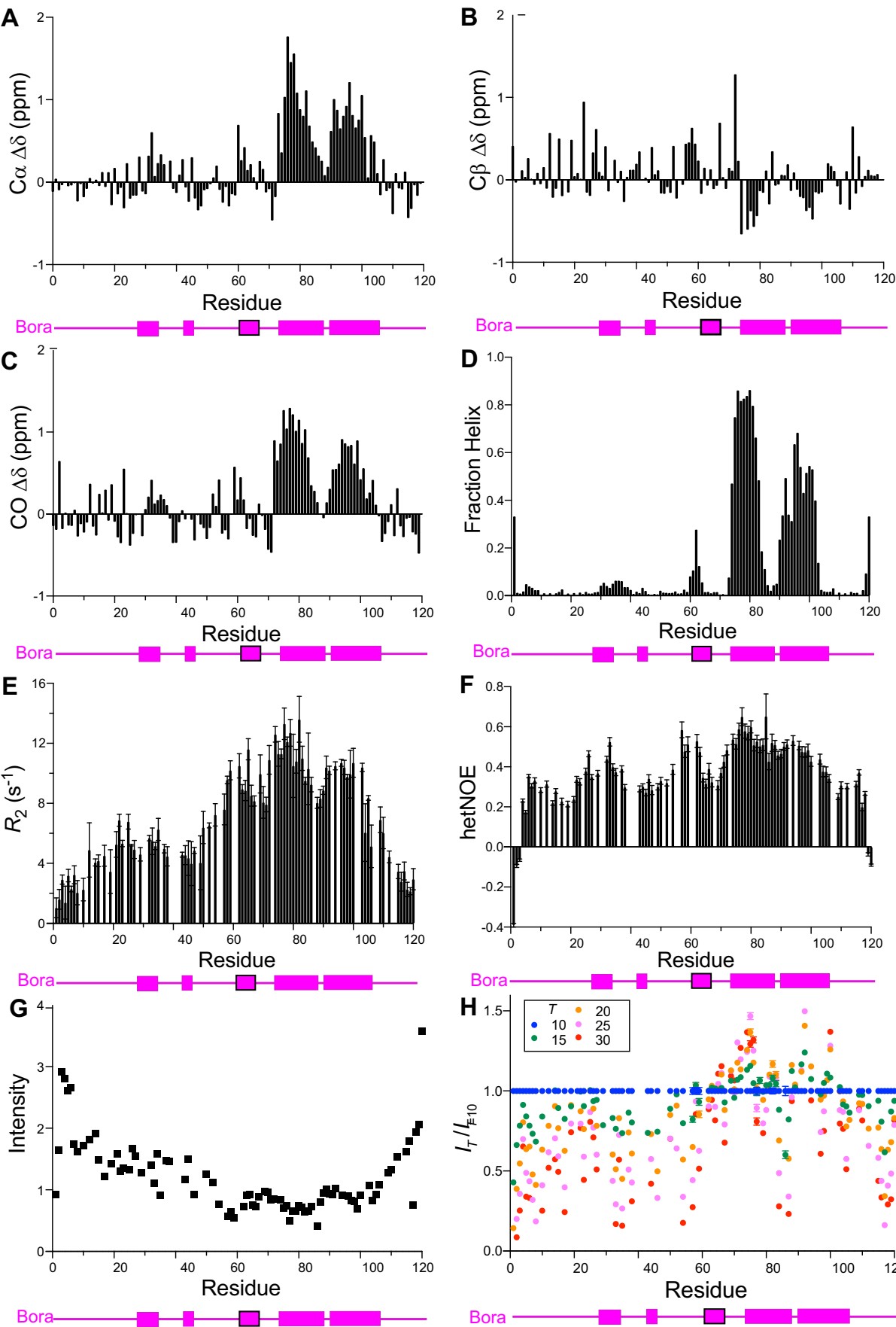

◀ **Figure EV3.  NMR characterisation of Bora 1–120.**

(**A–C**) Secondary shifts (Δδ) for Cα (**B**), CO (**C**) and Cβ (**D**) nuclei in Bora 1–120. Coil values are subtracted from measured shift values to give secondary shifts ($\Delta\delta = \delta - \delta_{ref}$). A representation of the model-predicted Bora secondary structure (in complex with Aurora-A or Aurora-A/PLK1) is shown below in pink, with helices represented as rectangles. The helix suggested to interact with the PLK1 'FW' pocket is bordered in black. (**D**) Residue-specific helical propensities within Bora 1–120 calculated using a TALOS-N analysis of all backbone chemical shifts. The analysis indicates two regions of high helical propensity towards the C-terminus. A representation of the model-predicted Bora secondary structure is shown below in pink, with helices represented as rectangles. The helix suggested to interact with the PLK1 'FW' pocket is bordered in black. (**E**) Transverse relaxation rates ($R_2$) for backbone $^{15}$N nuclei in Bora 1–120. A representation of the model-predicted Bora secondary structure is shown below in pink, with helices represented as rectangles. The helix suggested to interact with the PLK1 'FW' pocket is bordered in black. (**F**) Heteronuclear NOE values for HN groups in Bora 1–120. A representation of the model-predicted Bora secondary structure is shown below in pink, with helices represented as rectangles. The helix suggested to interact with the PLK1 'FW' pocket is bordered in black. (**G**) Raw peak intensities within $^{1}$H-$^{15}$N HSQC spectrum at 10 °C for Bora 1–120 (mirrors the $R_2$ result). A representation of the model-predicted Bora secondary structure is shown below in pink, with helices represented as rectangles. The helix suggested to interact with the PLK1 'FW' pocket is bordered in black. (**H**) Peak intensity ratios to compare the change in intensities with increasing temperature. A representation of the model-predicted Bora secondary structure is shown below in pink, with helices represented as rectangles. The helix suggested to interact with the PLK1 'FW' pocket is bordered in black. Source data are available online for this figure.

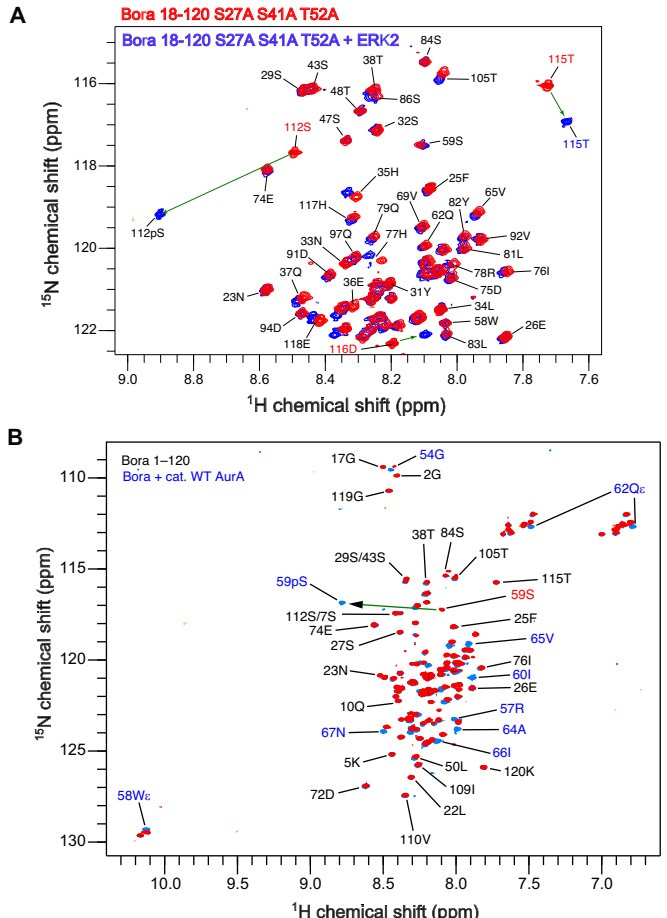

**Figure EV4. Phosphorylation of Bora by ERK2 and Aurora-A.**

(A) $^1$H-$^{15}$N HSQC spectra of Bora 18–120 S27A S41A T52A before (red) and after incubation with ERK (blue) in the presence of ATP/Mg$^{2+}$. A significant downfield $^1$H shift is seen for the Ser112 peak when phosphorylated, as well as small changes to neighbouring residues. (B) $^1$H-$^{15}$N HSQC spectra of Bora 1–120 before (red) and after incubation with Aurora-A kinase domain (blue) in the presence of ATP/Mg$^{2+}$. A significant downfield $^1$H shift is seen for the Ser59 peak when phosphorylated but note the unusual upfield $^{15}$N shift. Small changes to peak positions for neighbouring residues are also observed.

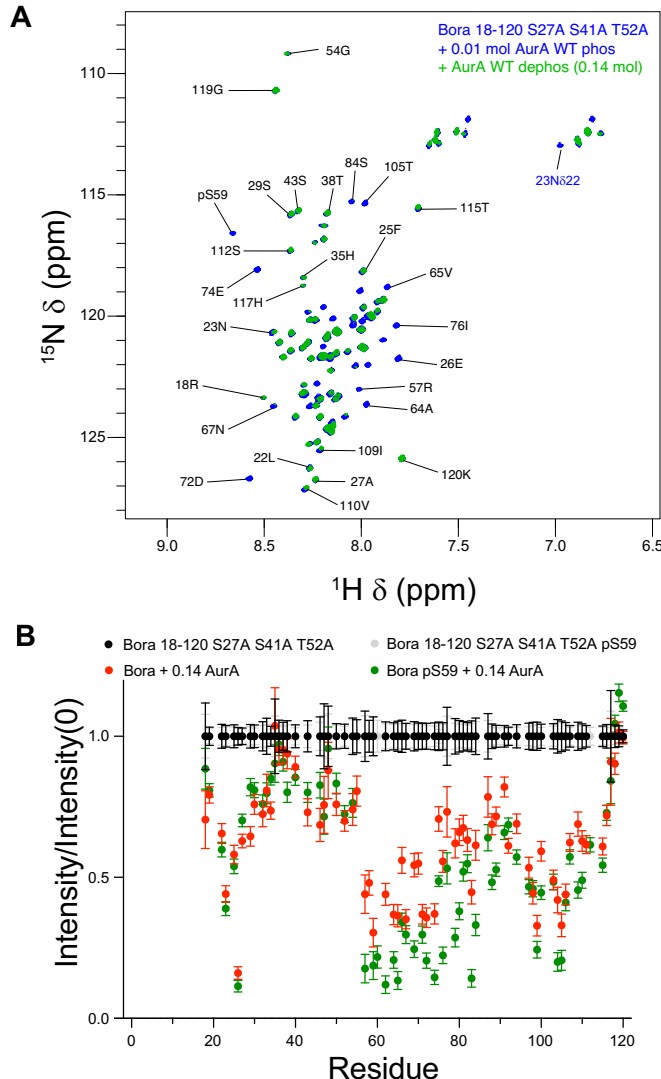

**Figure EV5.   Bora phosphorylation on Ser59 improves binding to Aurora-A.**

(A) $^1$H-$^{15}$N HSQC spectra for Aurora-A-phosphorylated Bora 18–120 S27A S41A T52A pS59 before (blue) and after addition of Aurora-A 122–403 (green). Bora peaks show sequence-specific attenuation in the presence of Aurora-A. (B) Change in peak intensity in the $^1$H-$^{15}$N HSQC spectrum of Bora 18–120 S27A S51A T52A with and without phosphorylation of Bora by Aurora-A at Ser59 (black—no phosphorylation, grey—with phosphorylation) and when Aurora-A is present (red—unphosphorylated Bora and Aurora-A, green—phosphorylated Bora with Aurora-A). The reduction in peak intensities around Ser59 (and the core sequence ~57–85) is higher for the phosphorylated version. Error bars ($\Delta R$) for peak intensity ratios ($R$) were determined from a measure of the baseline noise ($\Delta z$) in each spectrum, using $(\Delta R/R)^2 = (\Delta z_1/z_1)^2 + (\Delta z_2/z_2)^2$. Each spectrum was recorded from one sample ($n = 1$). Source data are available online for this figure.

