## [Peer Review File · EMBO Reports]

Bora Bridges Aurora-A Activation and Substrate Recognition of PLK1

Jennifer Miles, Matthew Batchelor, Martin Walko, Vanda Gunning, Andrew Wilson, Megan Wright, and Richard Bayliss

Corresponding author: Richard Bayliss (r.w.bayliss@leeds.ac.uk)

Review Timeline:

Transferred from Review Commons:	11th Nov 25
Editorial Decision:	27th Nov 25
Revision Received:	1st Dec 25
Editorial Decision:	9th Dec 25
Revision Received:	11th Dec 25
Accepted:	18th Dec 25

Transaction Report:

The logo for Review Commons, featuring the word "Review" in a large, blue, serif font with a diagonal slash through the letter 'v', and the word "COMMONS" in a smaller, blue, sans-serif font below it.

This manuscript was transferred to EMBO Reports following peer review at Review Commons.

Review #1**1. Evidence, reproducibility and clarity:****Evidence, reproducibility and clarity (Required)**

Miles et al. used a combination of AlphaFold modeling, biochemical assays of mutant constructs and NMR spectroscopy to model the ternary complex of Aurora A, Bora and Plk1, and elucidate how Bora can act as a molecular bridge that facilitates the phosphorylation of the activation loop Thr210 within Plk1 by Aurora A. Their studies identified an interaction between residues 52-73 within Bora and the 'FW' pocket on the N-terminal lobe of Plk1, which binds Phe56 and Trp58 of Bora. Additionally, Ser59 of Bora was identified as a good Aurora A substrate using a Bora peptide array, and pSer59 was predicted to form bridging interactions with Aurora Arg205 and Plk1 Arg59. This was supported by NMR and biochemical assays. In addition, the authors validate that phosphorylation of Ser-112 on Bora enhances stabilization of the Aurora A-Bora complex. Overall, the model revealed novel details of the interactions within the Aurora A-Bora-Plk1 ternary complex that are supported by the biochemical and NMR data. The work will be of significant interest to basic scientists whose work involves protein kinase signaling, cell division/mitosis, signal transduction, and cancer biology. We recommend publication of this manuscript with the following minor changes and additions.

1. In the introduction, on page 2, the authors seem a little confused about the Plk1 Polo-box domain - text as written: "...kinase domain linked to tandem Polo-box domains (PBD)", and cite a review paper. Actually, there is only a single Polo-box domain in these kinases, which contains both Polo-boxes and a bit of the upstream linker region. The "PBD" terminology denotes this 2-Polo-box +linker structure. Perhaps it would be better here to cite the PBD structure (Elia et al., Cell, 2002) as a primary citation here.
2. Similarly, the line "...during the G2/M transition following successful DNA damage repair" cites the Seki et al paper, but those findings are shown in the Macurek et al paper, not the Seki et al paper.
3. Using the model of the ternary complex as shown in Figure 1B, deletion constructs of Bora missing regions within the disordered loops, but still retaining the residues that bind the PBD, FW pocket and Aurora A, can be modeled and tested to see if such deletions can improve the ipTM scores and binding affinity.
4. On page 5, "S112A" within the sentence "Unexpectedly, the F56A/W58A Bora was less efficiently phosphorylated on S112A (Supplementary Figure S11, F compared to H and Supplementary Table S4)." This should be "S112".
5. In the assays shown in Figure 2D, the presence of excess F56AW58A Bora that remained

unphosphorylated on S112 may complicate the interpretation of the results. Can the authors show that the S112-phosphorylated F56AW68A Bora is predominantly bound to Aurora A in such a mixture, perhaps by NMR using labelled pS112 F56AW58A Bora and unlabeled S112 F56AW58A Bora?

6. Please expand Figure 3A to better show the FW pocket-forming residues on Plk1.

7. It would be helpful to label the peaks in the mass spectra in Fig. S11 with the phospho-species that they correspond to.

8. In the last paragraph on page 7, "see we" in the sentence "As well as a decrease in intensity around pSer112 in Bora, see we an overall effect with decreased intensity across most of the Bora sequence." Should be corrected to "we see".

9. While not required, it would be helpful if binding of Bora to Aurora A after Erk2 phosphorylation could be shown using fluorescence polarization or ITC to lend additional support to the NMR data for S112 and S59 phosphorylation and for CEP192 and TPX2 competition.

10. The Aurora A phosphorylation motif has been further defined beyond that reported by the Pinna lab in 2005. Notably, the Ser-59 sequence on Bora (F-R-W-S-I), has, in addition to dominant selection for AR in the -2 position, both favorable -1 (W) and +1 (I) positions based on peptide library measurements (Alexander et al., Science Signaling 2011), further arguing that it may be an excellent Aurora A phosphorylation site.

11. Have the authors tried to model the *Drosophila melanogaster* Aurora A-Bora-Polo complex to see if the Asn substitution of Bora Ser59, and the expected loss of the interactions between Bora pSer59 and Plk1 Arg59 and Aurora A Arg205 are compensated by other features?

12. Given the relevance of the recent publication from Zhu et al. in <https://doi.org/10.1038/s41467-025-63352-y> to this study, the authors may want to comment on, or test, the relative importance of PKA and Aurora A as a potential kinase for Bora S59. While those authors argue that PKA phosphorylates Bora on Ser-59, one could easily imagine a model in which either PKA or Aurora A could initially phosphorylate that site followed by a propagation step after initial Aurora A activation, in which Aurora A phosphorylation of Bora Ser-59 is the dominant process.

-Dan Lim and Michael Yaffe

2. Significance:

Significance (Required)

The work is well done and clearly presented.

3. How much time do you estimate the authors will need to complete the suggested revisions:

Estimated time to Complete Revisions (Required)

(Decision Recommendation)

Between 1 and 3 months

4. Review Commons values the work of reviewers and encourages them to get credit for their work. Select 'Yes' below to register your reviewing activity at Web of Science Reviewer Recognition Service (formerly Publons); note that the content of your review will not be visible on Web of Science.

Yes

Review #2

1. Evidence, reproducibility and clarity:

Evidence, reproducibility and clarity (Required)

****Summary:****

PLK1 is one of the master regulators of cell division. The activation of PLK1 requires the activation loop phosphorylation at T210, mediated by Aurora A kinase. However, Aurora A phosphorylation of PLK1 T210 requires Bora, one of the several activators of Aurora A kinase. While the molecular requirement of Aurora A kinase and Bora for PLK1 activation is well established, the mechanistic understanding of how Bora facilitates PLK1 activation by Aurora A has remained an important open question for a long time. Exploiting the latest development in AI-driven structure prediction, three independent studies provide a structural and mechanistic basis for PLK1 activation by Aurora A and Bora. Here, Miles et al. have generated AlphaFold models, further characterised some of the interfaces using NMR, and validated the contribution of intermolecular interactions at suggested interfaces in vitro using recombinant proteins in kinase assays. Overall, this is a well-executed work providing important new insights into our understanding of the activation of the critical regulator of cell division, PLK1. However, as the authors have highlighted in the discussion section, one limitation of this modelling study is that the models still do not entirely explain how these interactions facilitate the phosphorylation of Thr210, as this residue is oriented far away from Aurora A's active site for the reaction to take place. Despite this

limitation, I believe this is an important work that advances our understanding significantly.

****Comments:****

Experimental data satisfactorily support claims. Hence, most of my comments are minor in nature.

Points to consider during revision:

Page 5: '... a K82R PLK1 mutant was used to increase the stability of the protein' - It is not clear how this mutation confers increased stability of the protein. The authors do not show any data to support this. Isn't the PLK1 K82R an ATP-binding-deficient, kinase-inactive mutant?

All panels showing the Alphabridge diagram - it would be helpful if pictorial definitions of the colour codes were provided with corresponding score ranges (in addition to the description in the figure legend).

Fig 2B - The Fluorescence anisotropy assay curves do not reach a plateau. Though the effect of mutation on binding affinity is pretty clear, if possible, I suggest including more data points at higher concentrations and estimating apparent K_d values.

The cartoon representation of the structures and molecular interfaces - better to avoid shadows, as they compromise the clarity of the figures, particularly the ones where side chains are shown in stick representation.

It is important to discuss how the parallel studies by Verza et al. and Pillan et al. complement this study, highlighting similarities and differences.

2. Significance:

Significance (Required)

As highlighted in the summary, a mechanistic understanding of how PLK1 is activated by Aurora A kinase and its activator Bora has remained a long-standing open question. As PLK1 is one of the major regulators of cell division, which exerts its function (via phosphorylating numerous substrates) during different stages of mitosis, understanding its activation mechanism is of critical interest for those working on the cell cycle in general

and cell division in particular. A key limitation of this study is the lack of any cellular functional evaluation of the interaction interfaces.

3. How much time do you estimate the authors will need to complete the suggested revisions:

Estimated time to Complete Revisions (Required)

(Decision Recommendation)

Less than 1 month

4. Review Commons values the work of reviewers and encourages them to get credit for their work. Select 'Yes' below to register your reviewing activity at Web of Science Reviewer Recognition Service (formerly Publons); note that the content of your review will not be visible on Web of Science.

Yes

Review #3

1. Evidence, reproducibility and clarity:

Evidence, reproducibility and clarity (Required)

****Summary.****

Miles and co-workers have carried out a careful and high-quality study of the activation mechanisms of the mitotic kinase PLK1. Multiple proteins have been implicated in PLK1 activation and localisation as cell enter and pass through mitosis. Initial activation of PLK1 is promoted by a complex of Bora with another kinase Aurora A. Later in mitosis, this activated PLK1 associates with mitotic spindle and centrosome proteins regulating different aspects of mitosis and cytokinesis. In this study, Miles et al. extend previous work on this question by proposing and testing detailed models for Bora/Aurora A-mediated activation of PLK1 to elucidate the mechanism of this reaction.

Using the latest Alphafold they generate a series of models of the PLK1/Bora/Aurora A complex to home in on the key regions mediating interactions of the three proteins. This approach suggests an arrangement where the first ~120 amino acids of Bora wrap Aurora A and create an interaction surface for the N-terminal kinase domain of PLK1. This orients

Thr210 in PLK1 towards Aurora A creating a situation likely favourable for phosphorylation, although has the authors discuss there are some caveats to this. A further prediction of the modelling helps explain the requirement for Bora phosphorylation to promote the interaction with Aurora A. This data is presented in Fig. 1 and Fig. S1-S3.

In the subsequent figures the details of this model are tested using biochemical assays and structural biology methods to validate key predictions. First the PLK1 interaction with Bora was shown to require the conserved F/W motif of Bora and a conserved pocket close to R106 on PLK1 (Fig. 2 and 3). In reconstituted PLK1 activation assays the F/W motif mutant Bora showed greatly attenuated pThr210 phosphorylation. This reaction also required phosphorylation of Bora at S112, presumably due to the interaction with Aurora A. An R106A mutant PLK1 showed reduced binding to Bora and reduced kinase activation. This data is clear and provides compelling support for the model.

Using NMR the authors then investigate the interaction between Bora and Aurora A, and more specifically the requirement for Bora phosphorylation at Ser112. The NMR data in Fig. 4 and Fig. 6 provide good support for the AlphaFold model. A helpful comparison with known Aurora A binding proteins is also shown to highlight the way CEP192, TPX2 and TACC3 contact a series of conserved pockets on the surface of Aurora A which are common to the Bora interaction. S59 phosphorylation by Aurora A is also shown to play an important role in contacting PLK1 and is required for pThr210 phosphorylation.

In summary, the authors have made valuable progress in working out details of the PLK1 activation mechanism, that extends previous work in the field.

****Major comments.****

It would be helpful to measure the level of pThr210 PLK1 in some experiments and graph the data. The current presentation is Fig. 2D-E is qualitative rather than quantitative.

Have the authors measured the binding affinity of the F/W mutant Bora for PLK1 using the assay in Fig. 2B? Likewise, for Fig. 7 the S59 mutant could be tested to see if it affects PLK1 binding or activation.

It would be helpful if measurements of pThr210 PLK1 for all conditions were shown in the graph Fig. 7F.

****Minor comments.****

I found Figure S1B easier to understand than Fig S1A and Fig 1A-B. Some of the supplemental data Fig. S1C-E could be moved to a revised Figure 1, dropping the current Fig. 1A-B. Can the interaction plots (Fig. S1C-D) be rotated to have the same original at the top and order of proteins (i.e. Bora > Aurora A > {plus minus} PLK1 depending on the plot). Figure 3F. Typo "Strongyl" not "Strongly". Figure 3 could be supplemental material. Fig. 7E. Run a positive control reaction +ERK2 on the second gel to allow direct comparison of pThr210 across all the conditions tested.

2. Significance:

Significance (Required)

Timely and orchestrated activation of multiple mitotic protein kinases is crucial for the alignment and segregation of chromosomes, and for the process of cell division. In this study the authors explore how activation of the mitotic kinase PLK1 is triggered by another mitotic kinase Aurora A, and the role played by a scaffold protein Bora.

Strengths: Detailed analysis of mechanism using biochemical and structural approaches.

Limitations: The study is focussed on the biochemical and structural mechanisms rather than the cellular outcomes. Some data would benefit from additional quantitative measurement.

Relevance: Cancer and cell biology due to the role of Aurora A in many cancers.

Reviewer expertise: Biochemistry, molecular and cell biology.

3. How much time do you estimate the authors will need to complete the suggested revisions:

Estimated time to Complete Revisions (Required)

(Decision Recommendation)

Between 1 and 3 months

4. Review Commons values the work of reviewers and encourages them to get credit for their work. Select 'Yes' below to register your reviewing activity at Web of Science

Reviewer Recognition Service (formerly Publons); note that the content of your review will not be visible on Web of Science.

No

Full Revision

Manuscript number: RC-2025-03156

Corresponding author(s): Richard, Bayliss

1. General Statements [optional]

We appreciated the positive, detailed and helpful feedback from all three reviewers.

This section is mandatory. Please insert a point-by-point reply describing the revisions that were already carried out and included in the transferred manuscript.

Reviewer #1 (Evidence, reproducibility and clarity (Required)):

Miles et al. used a combination of AlphaFold modeling, biochemical assays of mutant constructs and NMR spectroscopy to model the ternary complex of Aurora A, Bora and Plk1, and elucidate how Bora can act as a molecular bridge that facilitates the phosphorylation of the activation loop Thr210 within Plk1 by Aurora A. Their studies identified an interaction between residues 52-73 within Bora and the 'FW' pocket on the N-terminal lobe of Plk1, which binds Phe56 and Trp58 of Bora. Additionally, Ser59 of Bora was identified as a good Aurora A substrate using a Bora peptide array, and pSer59 was predicted to form bridging interactions with Aurora Arg205 and Plk1 Arg59. This was supported by NMR and biochemical assays. In addition, the authors validate that phosphorylation of Ser-112 on Bora enhances stabilization of the Aurora A-Bora complex. Overall, the model revealed novel details of the interactions within the Aurora A-Bora-Plk1 ternary complex that are supported by the biochemical and NMR data. The work will be of significant interest to basic scientists whose work involves protein kinase signaling, cell division/mitosis, signal transduction, and cancer biology. We recommend publication of this manuscript with the following minor changes and additions.

1. In the introduction, on page 2, the authors seem a little confused about the Plk1 Polo-box domain - text as written: "...kinase domain linked to tandem Polo-box domains (PBD)", and cite a review paper. Actually, there is only a single Polo-box domain in these kinases, which contains both Polo-boxes and a bit of the upstream linker region. The "PBD" terminology denotes his 2-Polo-box +linker structure. Perhaps it would be better here to cite the PBD structure (Elia et al., Cell, 2002) as a primary citation here.

Thank you for finding this error, the text has been updated and the new citation included within the text on line 65.

2. Similarly, the line "...during the G2/M transition following successful DNA damage repair" cites the Seki et al paper, but those findings are shown in the Macurek et al paper, not the Seki et al paper.

Thank you for finding this error, the new citation included within the text on line 69.

3. Using the model of the ternary complex as shown in Figure 1B, deletion constructs of Bora missing regions within the disordered loops, but still retaining the residues that bind the PBD, FW pocket and Aurora A, can be modeled and tested to see if such deletions can improve the ipTM scores and binding affinity.

AlphaFold3 modelling was attempted with shorter regions of Bora to see the effect on the ipTM scores. Unfortunately, when Bora was reduced to shorter sequences, such as 18-88 or 18-45 modelled with 68-120, the models became inconsistent and of a low quality. Models were also created including the short region of Bora surrounding Ser252 that interacts with the polo box domain as well as Bora 18-120, but this had minimal effect on the calculated ipTM scores.

4. On page 5, "S112A" within the sentence "Unexpectedly, the F56A/W58A Bora was less efficiently phosphorylated on S112A (Supplementary Figure S11, F compared to H and Supplementary Table S4)." This should be "S112".

Thank you for spotting this, the error has been corrected.

5. In the assays shown in Figure 2D, the presence of excess F56A/W58A Bora that remained unphosphorylated on S112 may complicate the interpretation of the results. Can the authors show that the S112-phosphorylated F56A/W58A Bora is predominantly bound to Aurora A in such a mixture, perhaps by NMR using labelled pS112 F56A/W58A Bora and unlabeled S112 F56A/W58A Bora?

¹⁵N/¹³C labelled of Bora 18-120 F56A W58A was produced and assigned. We then phosphorylated a sample using ERK2, tracking with NMR, and when the reaction had progressed to a 50:50 mixture of pSer112 and Ser112 (based on peak intensities) the kinase activity was quenched by addition of EDTA to sequester Mg²⁺. This produced a solution containing both pS112 and unphosphorylated S112 Bora species with marker peaks in HSQC spectra that could be used to directly compare Aurora-binding to the two species. Aurora-A was introduced to the sample and the peak intensities were monitored. Although both species are affected, there is much greater peak loss from the pS112 related peaks than those for unphosphorylated S112. This indicates that Aurora-A still preferentially binds pS112 Bora over S112 Bora when the F56A W58A mutation is present. This data has been included in Supplementary Figure S11.

6. Please expand Figure 3A to better show the FW pocket-forming residues on Plk1.

Figure 3 has been amended to reduce the size of the sequence alignments so that 3A could be made slightly larger.

7. It would be helpful to label the peaks in the mass spectra in Fig. S11 with the phospho-species that they correspond to.

This information has been added to the mass spectra in Fig. S11 (now supplementary Figure S14) to make them easier to view.

8. In the last paragraph on page 7, "see we" in the sentence "As well as a decrease in intensity around pSer112 in Bora, see we an overall effect with decreased intensity across most of the Bora sequence." Should be corrected to "we see".

Thank you for spotting this, the error has been corrected.

9. While not required, it would be helpful if binding of Bora to Aurora A after Erk2 phosphorylation could be shown using fluorescence polarization or ITC to lend additional support to the NMR data for S112 and S59 phosphorylation and for CEP192 and TPX2 competition.

This question has been partially answered in previous work by Tavernier et al. (2021), who showed improved binding of Aurora-A to Bora after Erk phosphorylation (by SPR), and they used labelled-TPX2 for a series of competition FP assays in that and the recent parallel study (Pillan et al. 2025).

We made initial efforts to perform additional FP assays using longer sections of Bora with different phosphorylation states but without success (perhaps due to the multisite-binding nature of the Bora–Aurora interaction, and difficulties with directly expressing phosphorylated Bora). The revised manuscript now includes some additional NMR data to show improved Bora–Aurora-A interaction after phosphorylation at Ser59 (Supplementary Figure S12).

10. The Aurora A phosphorylation motif has been further defined beyond that reported by the Pinna lab in 2005. Notably, the Ser-59 sequence on Bora (F-R-W-S-I), has, in addition to dominant selection for AR in the -2 position, both favorable -1 (W) and +1 (I) positions based on peptide library measurements (Alexander et al., Science Signaling 2011), further arguing that it may be an excellent Aurora A phosphorylation site.

Thank you for highlighting this publication and how it further reinforces the likelihood of Ser59 being an effective substrate for Aurora-A, this should have been included in the original manuscript. This citation has now been included.

11. Have the authors tried to model the Drosophila melanogaster Aurora A-Bora-Polo complex

to see if the Asn substitution of Bora Ser59, and the expected loss of the interactions between Bora pSer59 and Plk1 Arg59 and Aurora A Arg205 are compensated by other features?

A ternary complex between the *Drosophila melanogaster* orthologues was modelled using AlphaFold3 (Uniprot code PLK1 (Q9VVR2 72-165), Aurora-A kinase (Q9VGF9) 151-411 and PLK1 (P52304 21-280)). This model was analysed using PDBe PISA to identify potential interactions between the three proteins, focusing on residues that are not conserved between the human and *Drosophila* sequences. From this model a potential salt bridge was identified between *Drosophila* Bora Lys120 and PLK1 Glu93 that would not occur in the human ternary complex given Lys120 is replaced with an asparagine. This could be an alternative (kinase-independent) method for improved Bora-PLK1 interaction. When comparing the Bora:Aurora-A side of the predicted interface and focusing on the short region of Bora in between Aurora-A and PLK1, there were no clear differences seen in the residues predicted to bind to Aurora-A. This modelling has been included in Supplementary Figure S10 C and D.

12. Given the relevance of the recent publication from Zhu et al. to this study, the authors may want to comment on, or test, the relative importance of PKA and Aurora A as a potential kinase for Bora S59. While those authors argue that PKA phosphorylates Bora on Ser-59, one could easily imagine a model in which either PKA or Aurora A could initially phosphorylate that site followed by a propagation step after initial Aurora A activation, in which Aurora A phosphorylation of Bora Ser-59 is the dominant process.

A brief discussion of this recent publication has been added to the discussion, highlighting the similarities between the two publications and the importance of pSer59, as well as suggesting that in *cellulo* this modification could be achieved via more than one pathway. We also include some additional NMR data to show improved Bora–Aurora-A interaction after phosphorylation at Ser59 (Supplementary Figure S12).

-Dan Lim and Michael Yaffe

Reviewer #1 (Significance (Required)):

The work is well done and clearly presented.

Reviewer #2 (Evidence, reproducibility and clarity (Required)):

Summary:

PLK1 is one of the master regulators of cell division. The activation of PLK1 requires the activation loop phosphorylation at T210, mediated by Aurora A kinase. However, Aurora A phosphorylation of PLK1 T210 requires Bora, one of the several activators of Aurora A kinase. While the molecular requirement of Aurora A kinase and Bora for PLK1 activation is well established, the mechanistic understanding of how Bora facilitates PLK1 activation by Aurora A has remained an important open question for a long time. Exploiting the latest development in AI-driven structure prediction, three independent studies provide a structural and mechanistic basis for PLK1 activation by Aurora A and Bora. Here, Miles et al. have generated AlphaFold models, further characterised some of the interfaces using NMR, and validated the contribution of intermolecular interactions at suggested interfaces *in vitro* using recombinant proteins in kinase assays. Overall, this is a well-executed work providing important new insights into our understanding of the activation of the critical regulator of cell division, PLK1. However, as the authors have highlighted in the discussion section, one limitation of this modelling study is that the models still do not entirely explain how these interactions facilitate the phosphorylation of Thr210, as this residue is oriented far away from Aurora A's active site for the reaction to take place. Despite this limitation, I believe this is an important work that advances our understanding significantly.

Comments:

Experimental data satisfactorily support claims. Hence, most of my comments are minor in nature.

Points to consider during revision:

Page 5: '... a K82R PLK1 mutant was used to increase the stability of the protein' - It is not clear how this mutation confers increased stability of the protein. The authors do not show any data to support this. Isn't the PLK1 K82R an ATP-binding-deficient, kinase-inactive mutant?

Thank you for spotting this, the text has been updated to clarify that this version of PLK1 was used as it is acting as a substrate in the *in vitro* assay as we didn't want to see any PLK1 activity within this assay.

All panels showing the AlphaBridge diagram - it would be helpful if pictorial definitions of the colour codes were provided with corresponding score ranges (in addition to the description in the figure legend).

The AlphaBridge images have been updated to include details about the pI-DDT scores each of the different colours refer to.

Fig 2B - The Fluorescence anisotropy assay curves do not reach a plateau. Though the effect of

Full Revision

mutation on binding affinity is pretty clear, if possible, I suggest including more data points at higher concentrations and estimating apparent K_d values.

The direct binding assay was repeated with a higher concentration of PLK1 in order to try and see a top plateau. This was successful and has been included in Figure 2B (shown in black). The measured K_d was $24 \pm 3 \mu\text{M}$.

The cartoon representation of the structures and molecular interfaces - better to avoid shadows, as they compromise the clarity of the figures, particularly the ones where side chains are shown in stick representation.

The structural images have been remade to remove the shadows and improve the clarity of the images.

It is important to discuss how the parallel studies by Verza et al. and Pillan et al. complement this study, highlighting similarities and differences.

References to these two publications and details on the similarities and differences seen are now included in the discussion.

Reviewer #2 (Significance (Required)):

Significance:

As highlighted in the summary, a mechanistic understanding of how PLK1 is activated by Aurora A kinase and its activator Bora has remained a long-standing open question. As PLK1 is one of the major regulators of cell division, which exerts its function (via phosphorylating numerous substrates) during different stages of mitosis, understanding its activation mechanism is of critical interest for those working on the cell cycle in general and cell division in particular. A key limitation of this study is the lack of any cellular functional evaluation of the interaction interfaces.

Reviewer #3 (Evidence, reproducibility and clarity (Required)):

Summary.

Miles and co-workers have carried out a careful and high-quality study of the activation mechanisms of the mitotic kinase PLK1. Multiple proteins have been implicated in PLK1 activation and localisation as cell enter and pass through mitosis. Initial activation of PLK1 is promoted by a complex of Bora with another kinase Aurora A. Later in mitosis, this activated PLK1 associates with mitotic spindle and centrosome proteins regulating different aspects of

mitosis and cytokinesis. In this study, Miles et al. extend previous work on this question by proposing and testing detailed models for Bora/Aurora A-mediated activation of PLK1 to elucidate the mechanism of this reaction.

Using the latest AlphaFold they generate a series of models of the PLK1/Bora/Aurora A complex to home in on the key regions mediating interactions of the three proteins. This approach suggests an arrangement where the first ~120 amino acids of Bora wrap Aurora A and create an interaction surface for the N-terminal kinase domain of PLK1. This orients Thr210 in PLK1 towards Aurora A creating a situation likely favourable for phosphorylation, although as the authors discuss there are some caveats to this. A further prediction of the modelling helps explain the requirement for Bora phosphorylation to promote the interaction with Aurora A. This data is presented in Fig. 1 and Fig. S1-S3.

In the subsequent figures the details of this model are tested using biochemical assays and structural biology methods to validate key predictions. First the PLK1 interaction with Bora was shown to require the conserved F/W motif of Bora and a conserved pocket close to R106 on PLK1 (Fig. 2 and 3). In reconstituted PLK1 activation assays the F/W motif mutant Bora showed greatly attenuated pThr210 phosphorylation. This reaction also required phosphorylation of Bora at S112, presumably due to the interaction with Aurora A. An R106A mutant PLK1 showed reduced binding to Bora and reduced kinase activation. This data is clear and provides compelling support for the model.

Using NMR the authors then investigate the interaction between Bora and Aurora A, and more specifically the requirement for Bora phosphorylation at Ser112. The NMR data in Fig. 4 and Fig. 6 provide good support for the AlphaFold model. A helpful comparison with known Aurora A binding proteins is also shown to highlight the way CEP192, TPX2 and TACC3 contact a series of conserved pockets on the surface of Aurora A which are common to the Bora interaction. S59 phosphorylation by Aurora A is also shown to play an important role in contacting PLK1 and is required for pThr210 phosphorylation.

In summary, the authors have made valuable progress in working out details of the PLK1 activation mechanism, that extends previous work in the field.

Major comments.

It would be helpful to measure the level of pThr210 PLK1 in some experiments and graph the data. The current presentation in Fig. 2D-E is qualitative rather than quantitative.

Graphs displaying the levels of pThr210 produced in the assay are now shown in Supplementary Figure S4.

Have the authors measured the binding affinity of the F/W mutant Bora for PLK1 using the assay in Fig. 2B? Likewise, for Fig. 7 the S59 mutant could be tested to see if it affects PLK1 binding or activation.

The direct binding assay has been repeated with the use of a FAM-Bora peptide that incorporates the F56A W58A mutation which shows reduced binding (Figure 2B, shown in blue). A version of the Bora peptide phosphorylated on Ser59 was also tested in the direct binding assay and this shows a similar affinity for PLK1 to the wild-type sequence (Figure 2B, shown in red compared to the wild-type shown in black).

It would be helpful if measurements of pThr210 PLK1 for all conditions were shown in the graph Fig. 7F.

This graph has been updated to include the levels of phosphorylation seen for PLK1 in all of the conditions tested.

Minor comments.

I found Figure S1B easier to understand than Fig S1A and Fig 1A-B. Some of the supplemental data Fig. S1C-E could be moved to a revised Figure 1, dropping the current Fig. 1A-B. Can the interaction plots (Fig. S1C-D) be rotated to have the same original at the top and order of proteins (i.e. Bora > Aurora A > {plus minus} PLK1 depending on the plot).

Figure 1 and S1 have been rearranged to hopefully make them easier to understand, with all AlphaFold3 models of the full-length sequences kept in the supplementary figure and the focus in 1B just on the truncated model. The AlphaBridge plots have been rotated as suggested.

Figure 3F. Typo "Strongyl" not "Strongly".

Thank you for spotting this, this has been corrected in the updated manuscript.

Figure 3 could be supplemental material.

Thank you for your suggestion, but we have decided to keep this as a main figure.

Fig. 7E. Run a positive control reaction +ERK2 on the second gel to allow direct comparison of pThr210 across all the conditions tested.

These samples have been rerun on the same membrane and the levels of phosphorylation have been quantified and included in Figure 7F.

Reviewer #3 (Significance (Required)):

Timely and orchestrated activation of multiple mitotic protein kinases is crucial for the alignment and segregation of chromosomes, and for the process of cell division. In this study the authors explore how activation of the mitotic kinase PLK1 is triggered by another mitotic kinase Aurora A, and the role played by a scaffold protein Bora.

Full Revision

Strengths: Detailed analysis of mechanism using biochemical and structural approaches.

Limitations: The study is focussed on the biochemical and structural mechanisms rather than the cellular outcomes. Some data would benefit from additional quantitative measurement.

Relevance: Cancer and cell biology due to the role of Aurora A in many cancers.

Reviewer expertise: Biochemistry, molecular and cell biology.

Dear Prof. Bayliss

Thank you for the submission of your research manuscript to our journal. I have asked former referee #3 to have a look at your revised manuscript and point-by-point response. The referee has now provided his/her report (copied below) and supports publication after a minor textual revision. Please address the remaining concern also in a point-by-point response.

Before we can proceed with official acceptance I need you to format your manuscript according to our guidelines. Below my signature and the referee report you will find the general formatting instructions. In order to speed up the process I list here a few specific items that we will need and a few things that are often overlooked:

- 1) Source data. I sent you an e-mail with instructions on the 13th. We need unmodified raw data, organized in one folder per figure containing subfolders for each panel. The Figure folder is then uploaded as a ZIP file. Skimming over your figures, these seem mostly Western blots and quantifications. Source data for the main figures is mandatory, but optional for any supplementary material.
- 2) Data availability section (at the end of Methods): here we need access information for all data deposited on external repositories. We need the name of the database, the identifier and an URL that links directly to the dataset (not just the database). Mass spectrometry data should also be deposited in a repository. (see point 7 below).
- 3) We need an Author Checklist (point 4) and a Reagents and Tools table (point 12). The Reagents and Tools table needs to be uploaded as separate file, not be part of the manuscript text.
- 4) The author contributions are exclusively specified in the online manuscript submission system, not in the manuscript text. This text needs to be removed.
- 5) The information on funding in the Acknowledgments section must be congruent with that in the online submission system. Please make sure to add all funding information in the system.
- 6) Figure legends: Whenever you have quantification you need to specify the number of replicates and their nature (biological, technical), the statistical test used and the exact p-value. Figure 7F, S4, S11D: you need to show the individual data points in addition to the mean and error bar. The bar and error bars must be defined.
- 7) Supplementary information is called "Appendix" with Appendix Figure S# and Appendix Table S#. The table of content needs page numbers. Table S1 could be part of the Reagents and Tools table.
- 8) Please make sure to cite the two related papers.
- 9) We need five keywords on the title page.
- 10) The Conflict of Interest Statement is called Disclosure and Competing Interests Statement.
- 11) We need a small synopsis image that will be displayed on our webpage. Be aware that the final size is only 550 pixels width. Please have a look at this final small version and make sure that all text is legible.
- 12) We need a short summary (2 sentences) and a few bullet points describing your key findings.

I think these are the most important points.

Once your revised manuscript has been submitted, our editorial assistants will check for all points listed above, a team of data editors will check figure legends and Data Availability section and we will perform a routine image integrity screen.

Please let me know if anything is unclear or if you have any further questions.

I am looking forward to receiving your revised manuscript.

Kind regards,

Martina

Martina Rembold, PhD
Senior Editor

=====

Referee #1:

This is a timely study analysing the mechanism of action of Bora, which plays an essential role directing Aurora A mediated activation of PLK1 during entry into mitosis. In the revised manuscript the authors have carefully addressed the comments of the reviewers with revisions to the text and figures, and added new data on key points. The related studies from the Pintard and Musacchio groups are now mentioned.

One very minor point:

Lines 490-493: "Crystal structures of the chromosome passenger complex protein INCENP bound to the Aurora-B and C kinase domains show that it too wraps around the N-lobe, and phosphorylation of Ser893 and Ser894 in the INCENP sequence is vital for the complete stimulation of kinase activity (Abdul Azeez et al, 2019; Elkins et al, 2012)."

The Aurora C structure used for Figure S13A has bound inhibitor shown in the figure, so this should be mentioned in the text. I wasn't entirely sure why a drug-free structure for Aurora B (the dominant partner for INCENP in cells) wasn't used for the figure.

=====

GENERAL FORMATTING INSTRUCTIONS

2) individual production quality figure files as .eps, .tif, .jpg (one file per figure).

Please download our Figure Preparation Guidelines (figure preparation pdf) from our Author Guidelines pages <https://www.embopress.org/page/journal/14693178/authorguide> for more info on how to prepare your figures.

4) a complete author checklist, which you can download from our author guidelines (). Please insert information in the checklist that is also reflected in the manuscript. The completed author checklist will also be part of the RPF.

5) Please note that all corresponding authors are required to supply an ORCID ID for their name upon submission of a revised manuscript (). Please find instructions on how to link your ORCID ID to your account in our manuscript tracking system in our Author guidelines

()

6) We replaced Supplementary Information with Expanded View (EV) Figures and Tables that are collapsible/expandable online. A maximum of 5 EV Figures can be typeset. EV Figures should be cited as "Figure EV1, Figure EV2" etc... in the text and their respective legends should be included in the main text after the legends of regular figures.

7) Before submitting your revision, primary datasets (and computer code, where appropriate) produced in this study need to be deposited in an appropriate public database (see < <https://www.embopress.org/page/journal/14693178/authorguide#dataavailability>>).

The accession numbers and database should be listed in a formal "Data Availability" section (placed after Materials & Method) that follows the model below (see also < <https://www.embopress.org/page/journal/14693178/authorguide#dataavailability>>). Please note that the Data Availability Section is restricted to new primary data that are part of this study.

Data availability

Additional information on source data and instruction on how to label the files are available

10) Figure legends and data quantification:

- the name of the statistical test used to generate error bars and P values,
 - the EXACT p-values,
 - the number (n) of independent experiments (please specify technical or biological replicates) underlying each data point,
 - the nature of the bars and error bars (s.d., s.e.m.)
-
- If the data are obtained from n {less than or equal to} 5, show the individual data points in addition to the SD or SEM.
 - If the data are obtained from n {less than or equal to} 2, use scatter blots showing the individual data points.

11) Our journal encourages inclusion of *data citations in the reference list* to directly cite datasets that were re-used and obtained from public databases. Data citations in the article text are distinct from normal bibliographical citations and should directly link to the database records from which the data can be accessed. In the main text, data citations are formatted as follows: "Data ref: Smith et al, 2001" or "Data ref: NCBI Sequence Read Archive PRJNA342805, 2017". In the Reference list, data citations must be labeled with "[DATASET]". A data reference must provide the database name, accession number/identifiers and a resolvable link to the landing page from which the data can be accessed at the end of the reference. Further instructions are available at .

12) All Materials and Methods need to be described in the main text using our 'Structured Methods' format. According to this format, the Methods section includes a Reagents and Tools Table (listing key reagents, experimental models, software and relevant equipment and including their sources and relevant identifiers) followed by a Methods and Protocols section describing the methods, ideally using a step-by-step protocol format. The aim is to facilitate adoption of the methodologies across labs. Please download and fill our Reagents and Tools Table template (.docx), which you can find in our author guidelines: <https://www.embopress.org/page/journal/14693178/authorguide#structuredmethods>.

13) As part of the EMBO publication's Transparent Editorial Process, EMBO Reports publishes online a Review Process File to accompany accepted manuscripts. This File will be published in conjunction with your paper and will include the referee reports, your point-by-point response and all pertinent correspondence relating to the manuscript.

=====

Referee #1:

This is a timely study analysing the mechanism of action of Bora, which plays an essential role directing Aurora A mediated activation of PLK1 during entry into mitosis. In the revised manuscript the authors have carefully addressed the comments of the reviewers with revisions to the text and figures, and added new data on key points. The related studies from the Pintard and Musacchio groups are now mentioned.

One very minor point:

Lines 490-493: "Crystal structures of the chromosome passenger complex protein INCENP bound to the Aurora-B and C kinase domains show that it too wraps around the N-lobe, and phosphorylation of Ser893 and Ser894 in the INCENP sequence is vital for the complete stimulation of kinase activity (Abdul Azeez et al, 2019; Elkins et al, 2012)."

The Aurora C structure used for Figure S13A has bound inhibitor shown in the figure, so this should be mentioned in the text. I wasn't entirely sure why a drug-free structure for Aurora B (the dominant partner for INCENP in cells) wasn't used for the figure.

Rev_Com_number: RC-2025-03156

New_manu_number: EMBOR-2025-63171V1-T

Corr_author: Bayliss

Title: Bora Bridges Aurora-A Activation and Substrate Recognition of PLK1

Rev_Com_number: RC-2025-03156

New_manu_number: EMBOR-2025-63171V1-T

Corresponding author(s): Richard, Bayliss

Final round of review – November 2025

Referee #1: (former referee #3)

This is a timely study analysing the mechanism of action of Bora, which plays an essential role directing Aurora A mediated activation of PLK1 during entry into mitosis. In the revised manuscript the authors have carefully addressed the comments of the reviewers with revisions to the text and figures, and added new data on key points. The related studies from the Pintard and Musacchio groups are now mentioned.

One very minor point:

Lines 490-493: "Crystal structures of the chromosome passenger complex protein INCENP bound to the Aurora-B and C kinase domains show that it too wraps around the N-lobe, and phosphorylation of Ser893 and Ser894 in the INCENP sequence is vital for the complete stimulation of kinase activity (Abdul Azeez et al, 2019; Elkins et al, 2012)."

The Aurora C structure used for Figure S13A has bound inhibitor shown in the figure, so this should be mentioned in the text. I wasn't entirely sure why a drug-free structure for Aurora B (the dominant partner for INCENP in cells) wasn't used for the figure.

We have amended the text to clarify that the structure shown has an ATP-competitive inhibitor bound. "The structure of INCENP bound to Aurora-C and an ATP-competitive inhibitor BRD-7880 (PDB 6GR8) was aligned.."

Manuscript number: EMBOR-2025-63171V2

Title: Bora Bridges Aurora-A Activation and Substrate Recognition of PLK1

Author(s): Jennifer Miles, Matthew Batchelor, Martin Walko, Vanda Gunning, Andrew Wilson, Megan Wright, and Richard Bayliss

Dear Richard,

Thank you for the submission of your revised manuscript to EMBO reports.

We have checked everything and noticed a few remaining issues that would need your attention before we can proceed with formal acceptance:

1) Our editorial policies do not allow to base statements on 'data not shown'. With this in mind I note that the conclusion on page 7 is based on such a statement: "By contrast, and as observed previously (Tavernier et al, 2021), titrating PLK1 into labelled Bora resulted in very little by way of spectral change even at higher molar ratios (data not shown), highlighting the transient nature of the Bora/PLK1 interaction."

Please either show the data or remove/modify the statement.

2) Appendix: the figures and legends are missing the word "Appendix" throughout the file. Please relabel these as Appendix Figure S# and Appendix Table S#. The nomenclature in the text seems correct.

3) During our routine image checks, we noticed that some figures have rather low resolution. Please upload Figure 2, Figure 7 and Appendix Figure S3 at higher resolution.

4) Statistics and figure legends:

Please add information related to n in the legends of figures 1C, E, F; 2B, 4B, E, 6A-C; EV5 B

- Please note that $n=2$ in figures EV2 A, B. Please remove any statistical analysis in this case.

- Please define the error bars in the legends of figures 4B, E, 6A-C; EV5 B

- Please note that the measure of center for the error bars needs to be defined in the legend of figure 7F

- Please provide the exact p values in the legend of figure 7F (unless $p < 0.0001$)

5) Please add MW markers for the Western blots shown.

6) I attach here the clean manuscript file with a modification introduced by me: preprint citations in the text need the prefix "preprint: "

7) Regarding preprints: please cite the related manuscripts, Esposito Verza et al, 2025 and Pillan et al, 2025 as "The EMBO Journal, in press"

This should be all we need.

Once you have made these minor revisions, please use the following link to submit your corrected manuscript:

Link Not Available

If all remaining corrections have been attended to, you will then receive an official decision letter from the journal accepting your manuscript for publication in the next available issue of EMBO reports. This letter will also include details of the further steps you need to take for the prompt inclusion of your manuscript in our next available issue.

Thank you for your contribution to EMBO reports.

Best regards,

Martina

The authors addressed the remaining editorial issues.

Prof. Richard Bayliss
University of Leeds
School of Molecular and Cellular Biology
Woodhouse Lane
Leeds LS2 9JT
United Kingdom

Dear Richard,

I am very pleased to accept your manuscript for publication in the next available issue of EMBO reports. Thank you for your contribution to our journal.

You may qualify for financial assistance for your publication charges - either via a Springer Nature fully open access agreement or an EMBO initiative. Check your eligibility: <https://link.springer.com/journal/44319/how-to-publish-with-us>

Kind regards,

Martina

>>> Please note that it is EMBO Reports policy for the transcript of the editorial process (containing referee reports and your response letter) to be published as an online supplement to each paper. If you do NOT want this, you will need to inform the Editorial Office via email immediately. More information is available here: <https://link.springer.com/partners/embo-press/editorial-policies#Peer%20review>

Rev_Com_number: RC-2025-03156
New_manu_number: EMBOR-2025-63171V3
Corr_author: Bayliss
Title: Bora Bridges Aurora-A Activation and Substrate Recognition of PLK1